# Learnable Sparsity for Vision Generative Models

**Yang Zhang**[*][♠]   **Er Jin**[*][◇]   **Wenzhong Liang**[*][♠]   **Yanfei Dong**[♠]   **Ashkan Khakzar**[▲]
**Philip Torr**[▲]   **Johannes Stegmaier**[◇]   **Kenji Kawaguchi**[♠]

[♠]National University of Singapore   [◇]RWTH Aachen University   [▲]University of Oxford

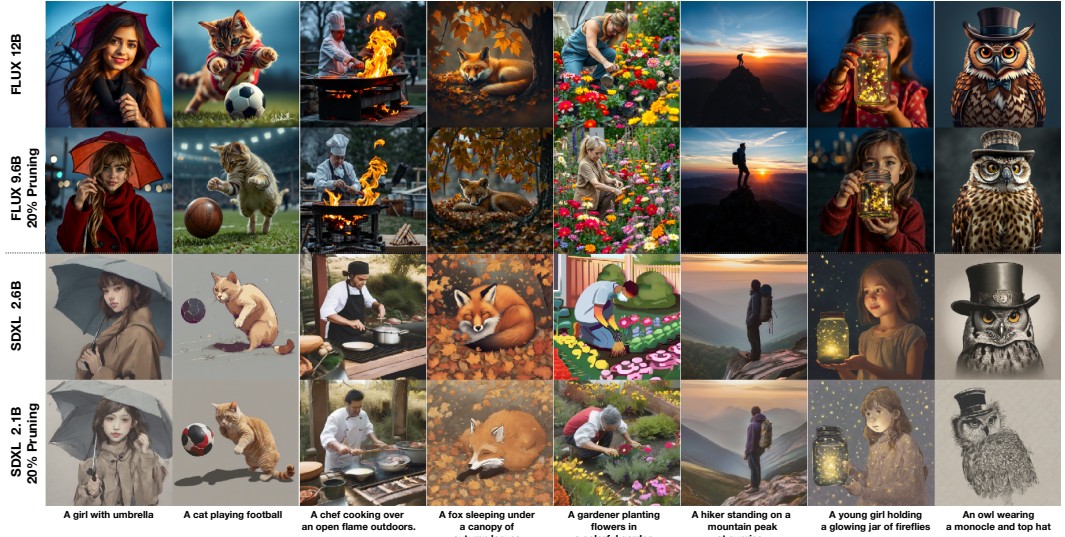

Figure 1: We present a differentiable masking approach designed for pruning vision generative models. Our method is highly efficient, achieving 20% sparsity in state-of-the-art diffusion and flow matching models using a small calibration set of 100 samples and only 10 A100 GPU hours. The mask learning phase requires only 50 optimization steps.

## Abstract

Generative models have achieved impressive advancements in various vision tasks. However, these gains often rely on increasing model size, which raises computational complexity and memory demands. The increased computational demand poses challenges for deployment, elevates inference costs, and impacts the environment. While some studies have explored pruning techniques to improve the memory efficiency of diffusion models, most existing methods require extensive retraining to maintain model performance. Retraining a large model is extremely costly and resource-intensive, which limits the practicality of pruning methods. In this work, we achieve low-cost pruning by proposing a general pruning framework for vision generative models that learns a differentiable mask to sparsify the model. To learn a mask that minimally deteriorates the model, we design a novel end-to-end pruning objective that spans the entire generation process over all steps. Since end-to-end pruning is memory-intensive, we further design a time step gradient checkpointing technique for the end-to-end pruning, a technique that significantly reduces memory usage during optimization, enabling end-to-end pruning within a limited memory budget. Results on the state-of-the-art U-Net diffusion models Stable Diffusion XL (SDXL) and DiT flow models (FLUX) show that our method efficiently prunes 20% of parameters in just 10 A100 GPU hours, outperforming previous pruning approaches.

---

[*]Equal contribution

# 1 INTRODUCTION

Recently, generative models, such as diffusion models and flow matching models, have made remarkable progress in various vision tasks, including text-to-image generation (Brooks et al., 2023; Rombach et al., 2022; Ramesh et al., 2022; Nichol et al., 2022), image inpainting (Saharia et al., 2022a; Lugmayr et al., 2022; Corneanu et al., 2024), super-resolution (Saharia et al., 2022b; Li et al., 2022), and video generation (Ho et al., 2022; Luo et al., 2023; Singer et al., 2022). This progress has been marked by the evolution of generative models toward increasingly larger sizes, transitioning from the U-Net-based Stable Diffusion 1 (SD1) (Rombach et al., 2022) to the larger SDXL (Podell et al., 2024), the transformer-based Stable Diffusion 3 (SD3), and more recently, the FLUX model (Esser et al., 2024; Black Forest Labs, 2024). The most recent FLUX model, with 12 billion parameters, is about 13 times larger than the SD2, which was developed just two years ago (Black Forest Labs, 2024). The rapid growth in size has introduced challenges in application: larger models demand larger GPUs and more computation during inference, limit deployment on smaller compute platforms, and substantially increase the carbon footprint.

Due to the potential challenges induced by larger models, prior methods have explored ways to reduce model size and computation, including model distillation (Meng et al., 2023; Huang et al., 2024) and model pruning (Fang et al., 2023). Compared to standard knowledge distillation that usually trains a student model from scratch, pruning is computationally more efficient. However, pruned models with fewer weights often suffer from degraded generative performance, making retraining necessary to restore their capabilities. Fang et al. (2023) estimate that diffusion model compression can demand 10% to 20% of the original training cost. Therefore, compressing an SD2 can consume up to $40,000$ GPU hours (Stability AI, 2022). Recently, Kim et al. (2024) showed that the cost of pruning SD2 can be reduced to around 300 A100 GPU hours with 0.22M calibration data. For pruning larger models, such as SDXL and FLUX, the retraining burden is more substantial.

One primary reason for the extensive retraining required in prior approaches is due to their coarse pruning criteria. Prior pruning approaches for diffusion models use simple heuristics or one-shot pruning strategies (Kim et al., 2024; Fang et al., 2023), which often fail to balance sparsity and performance. To improve the pruning selection criteria, differentiable masking has been explored in classical pruning research (Louizos et al., 2018; Wen et al., 2016; Liu et al., 2015; Han et al., 2016; Yuan & Lin, 2006) and has recently been adapted to large language models (LLMs) (Fang et al., 2024). However, several unique characteristics of vision generative models make the application of differentiable masking difficult. For instance, vision generative models are Markovian (Song et al., 2021a;b; Lipman et al., 2021), where each generation step only relies on the previous state. Consequently, a minor change at an intermediate generation step can cause a ripple effect that significantly distorts the final result.

In this work, we propose an end-to-end pruning scheme that adapts differentiable masking to vision generative models. With our innovative pruning scheme, we show for the first time, to the best of our knowledge, that a significant number of parameters can be removed using 10 A100 GPU hours and only 100 data samples, as illustrated in Figure 1. This approach has the advantage over the alternative of optimizing a mask in a per-step manner, which can introduce cumulative errors throughout the denoising steps and eventually lead to substantial quality degradation. Detailed discussion can be found in Section 4.

While prior works often require extensive retraining to recover model quality after pruning, we show that this step can be made far more lightweight. By combining pruning with targeted post-pruning adaptation, such as low-rank (LoRA) fine-tuning (Hu et al., 2022) or full-model retraining, our approach enables substantial quality recovery without prohibitive compute. This makes pruning practical even for large-scale diffusion models like SDXL, where retraining can otherwise be resource-intensive. However, end-to-end mask learning results in an extremely long gradient chain across all generation steps, leading to significant memory demands. For instance, performing end-to-end pruning on SDXL requires approximately 1400 GB of VRAM–equivalent to the capacity of 15 NVIDIA H100 GPUs. To address this, we adopt gradient checkpointing (Chen et al., 2016) and introduce time step gradient checkpointing. This approach reduces the VRAM usage for SDXL from 1400 GB to under 30 GB, with a minor overhead of an additional forward pass. An overview of end-to-end pruning and gradient checkpointing is shown in Figure 2.

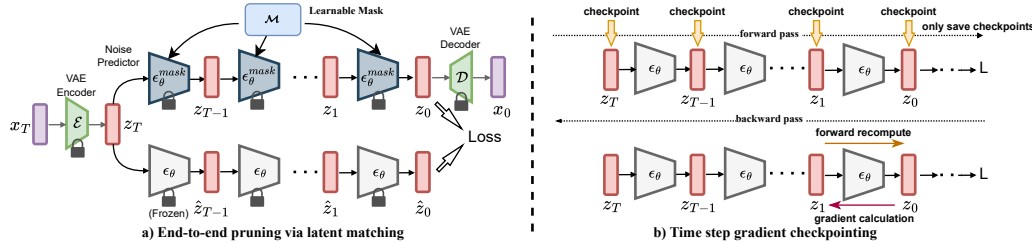

Figure 2: **Overview of end-to-end pruning framework and time step gradient checkpointing**. In a), end-to-end pruning learns a mask that applies to all denoising steps, thereby reducing model size while preserving the final denoised latent for semantic integrity. In b), only checkpoints are stored during the forward pass. During the backward pass, we first recompute the intermediate states between checkpoints at each step, then perform gradient calculation. Therefore, memory usage is reduced by $T$ times with only one additional forward pass.

We show that our method is general and scalable by pruning the most recent diffusion model (SDXL) and flow matching model (FLUX) in Section 6.2. Recent studies have shown that step distillation can greatly speed up inference by reducing generation steps (Salimans & Ho, 2022; Meng et al., 2023). However, step-distilled models are difficult to retrain due to their non-smoothness (Miao et al., 2024; Jia et al., 2024). We show that we can prune step-distilled models in Section 6.2 by pruning FLUX-schnell.

Overall, our contribution is summarized as follows: **(1)** We introduce **EcoDiff**, a model-agnostic, end-to-end structural pruning framework for vision generative models that learns a differentiable neuron mask, enabling efficient pruning across various architectures. **(2)** We develop a novel time step gradient checkpointing technique, significantly reducing memory requirements for end-to-end pruning to be feasible with lower computational resources. **(3)** We conduct extensive evaluations across U-Net diffusion models and diffusion transformers, demonstrating that our method can effectively prune 20% of model parameters without necessitating extensive retraining. Additionally, we show that our approach is compatible with step-distilled models and other acceleration methods. Moreover, EcoDiff supports lightweight post-pruning adaptation through both LoRA and full-model fine-tuning, enabling substantial quality recovery with minimal additional compute.

## 2 RELATED WORK

**Efficient diffusion models:** Several works focus on enhancing the efficiency of diffusion models at inference time. DDIM formulates diffusion processes with non-Markovian transformations, reducing the required diffusion steps for high-quality generation (Song et al., 2021a). The Latent Diffusion Model adopts a two-stage generation process, performing the diffusion process in latent space and decoding it back to the image domain using a Variational Autoencoder (VAE) (Rombach et al., 2022; Kingma & Welling, 2014). In addition, some prior works accelerate diffusion inference by sharing intermediate variables to reduce redundant computations (Ma et al., 2024b;a; So et al., 2023). However, these methods are often limited to specific architectures and do not generalize well to newer diffusion models incorporating multi-modal attention. Early stopping methods attempt to terminate the diffusion process once satisfactory generation is reached, but they can sacrifice fine-grained details (Lyu et al., 2022). Distillation-based approaches train a student model that is either smaller or can generate with fewer diffusion steps (Salimans & Ho, 2022; Meng et al., 2023; Gu et al., 2023; Hsiao et al., 2024). Nevertheless, model distillation requires extensive retraining.

**Diffusion training efficiency:** Beyond inference speedup, research also targets the efficiency of diffusion model training. These efforts range from data-centric acceleration, such as lossless speedup through unbiased dynamic data pruning (Qin et al., 2024) or approaches for data-efficient training via sample reweighting (Li et al., 2025), to architectural and objective acceleration that utilizes masked transformers for faster training (Zheng et al., 2024), patched denoising diffusion models for high-resolution synthesis (Ding et al., 2023), general fast diffusion model designs (Wu et al., 2023), and min-SNR weighting strategies (Hang et al., 2023). These approaches primarily target

training-time efficiency, whereas our work focuses on inference-time structural compression and is complementary to the above techniques.

**Model pruning:** Pruning reduces model size, thereby lowering both memory requirements for loading the model and computation demands during inference. Many classical pruning approaches learn a mask to prune CNN models (Louizos et al., 2018; Wen et al., 2016; Liu et al., 2015; Han et al., 2016; Yuan & Lin, 2006; Feng & Darrell, 2015). Recent advances in model pruning have predominantly focused on large language models (LLMs), where techniques such as unstructured and semi-structured pruning eliminate connections between neurons, and structured pruning targets neurons, attention heads, and layers (Kwon et al., 2022a; Fang et al., 2024; Zhang et al., 2024b;a; Yu et al., 2018; Ma et al., 2023; Kwon et al., 2022b; Sun et al., 2023). In contrast, pruning methods for diffusion models remain relatively underexplored. These approaches vary in their granularity: some involve structural pruning, while others focus on unstructured pruning. Some structural approaches (Castells et al., 2024; Li et al., 2023; Zhao et al., 2024) perform occlusion-based pruning that calculates an importance score for occluding a part through exhaustive search, but its high complexity limits scalability to complex pruning scenarios with many candidates. Unstructured pruning has also been explored, for example, Efficient Pruning of Text-to-Image Models (EPTI) (Ramesh & Zhao, 2024) uses criteria such as magnitude and WANDA, reporting strong results without demanding the extensive retraining common in many other pruning approaches. Fang et al. (2023) uses gradient information as a proxy for neuron importance. Compared to the occlusion method, it scales to more complex pruning cases. Other approaches (Kim et al., 2024) prune certain redundant blocks and recover the model with feature distillation to further reduce the pruning cost. Still, these methods require excessive retraining to retain the model performance.

## 3 PRELIMINARIES

**Iterative generative models.** Currently, SOTA generative models in vision are mostly iterative generative models, such as diffusion models and flow models. The sampling process in a latent diffusion model (LDM) iteratively reduces the noise in the initial noisy latent $z_T \sim \mathcal{N}(0, I)$, until reaching the final denoised latent $z_0$. This latent $z_0$ is then decoded via a pretrained decoder $\mathcal{D}$ to reconstruct an image $\hat{\mathbf{x}} = \mathcal{D}(z_0)$. One denoising step can be summarized by $f(z_t, y, t)$ as follows:

$$z_{t-1} = \frac{1}{\sqrt{\alpha_t}} \left( z_t - \frac{1 - \alpha_t}{\sqrt{1 - \bar{\alpha}_t}} \, \epsilon(z_t, t, y) \right) + \sigma_t \eta = f(z_t, y, t),$$

where $t \in \{T, T-1, \ldots, 1\}$. $\epsilon(z_t, t, y)$ represents the predicted noise conditioned on the previous latent $z_t$ and input condition $y$. The terms $\alpha_t$ and $\bar{\alpha}_t$ are noise schedule parameters. The term $\sigma_t$ denotes a controlled level of random noise, with $\eta \sim \mathcal{N}(0, \mathbf{I})$. Unlike diffusion models that model the generation process as a stochastic differential equation (SDE), flow models sample using an ordinary differential equation (ODE) that imitates a distribution flow. The sampling process exploits Euler method to approximates the integration:

$$X_{t+h} = X_t + h \cdot u_t(X_t, y), \tag{1}$$

where $X_t$ is the state of a particle in the flow at time step $t$, $u_t(X_t, y)$ is the neural network that predicts the velocity of the particle $X_t$, $h$ is the update step.

**Modules in transformer blocks.** Transformer blocks are one of the major building blocks in U-Net diffusion models (Rombach et al., 2022; Podell et al., 2024). In addition, recent diffusion transformers employ fully transformer-based architectures. This makes transformer blocks a primary target for pruning to improve efficiency. A transformer block consists of multi-head attention (MHA) and feed-forward network (FFN) (Vaswani et al., 2017). The MHA with $h$ attention heads is defined as,

$$\text{MHA}(Q, K, V) = (\text{attn}_1 \| \ldots \| \text{attn}_h) W^o, \tag{2}$$

where each $\text{attn}_i$ is a dot product attention for head $i$, which is computed as $\text{attn}_i = \text{softmax}\left( \frac{Q_i K_i^\top}{\sqrt{d_k}} \right) V_i$, with $Q_i = W_Q^i x$, $K_i = W_K^i x$, and $V_i = W_v^i x$ for an input $x$, $W_K^i \in \mathbb{R}^{d_{\text{model}} \times d_k}$, $W_Q^i \in \mathbb{R}^{d_{\text{model}} \times d_q}$, $W_V^i \in \mathbb{R}^{d_{\text{model}} \times d_v}$ and $W^O \in \mathbb{R}^{d_v \times d_{\text{model}}}$. Additionally, $\|$ denotes concatenation along the feature dimension across the attention heads. The FFN applied after MHA is defined as

$$\text{FFN}(x) = \sigma(x W_1 + b_1) W_2 + b_2, \tag{3}$$

where $W_1 \in \mathbb{R}^{d_{\text{model}} \times d_{\text{ff}}}$ and $W_2 \in \mathbb{R}^{d_{\text{ff}} \times d_{\text{model}}}$. $\sigma(\cdot)$ represents an activation function like GELU or GeGLU (Hendrycks & Gimpel, 2016; Shazeer, 2020).

---

**Algorithm 1** End-to-End Diffusion Model Pruning

---

**Input:** Pre-trained denoise model $\epsilon_\theta$ and masked pre-trained denoise model $\epsilon_\theta^{\text{mask}}$ with masking parameters $\boldsymbol{\mathcal{M}}$ with initial value $\boldsymbol{\mathcal{M}}_{\text{init}}$, text prompts $x \sim \mathcal{X} = \{x^i\}_{i=1}^N$, regularization coefficient $\beta$, learning rate $\eta$
**Output:** Learned pruning mask $\boldsymbol{\mathcal{M}}$
1: $z_T \sim \mathcal{N}(0, \mathbf{I})$          ▷ Initialize with latent random noise
2: **for** $x$ **in** $\mathcal{X}$ **do**          ▷ $x$ as text prompt for training
3:      $z_0 \leftarrow \mathcal{F}_{\epsilon_\theta}(z_t, x)$          ▷ Original latent $z_0$ via $\epsilon_\theta$
4:      $\hat{z}_0 \leftarrow \mathcal{F}_{\epsilon_\theta^{\text{mask}}}(z_t, x, \boldsymbol{\mathcal{M}})$          ▷ Masked latent $\hat{z}_0$ via $\epsilon_\theta^{\text{mask}}$
5:      $\mathcal{L} = \sum_x \|\hat{z}_0 - z_0\|_2 + \beta\|\boldsymbol{\mathcal{M}}\|_0$          ▷ Loss with regularization
6:      $\boldsymbol{\mathcal{M}} \leftarrow \boldsymbol{\mathcal{M}} + \eta\frac{d\mathcal{L}}{d\boldsymbol{\mathcal{M}}}$
7: **end for**
8: **return** $\boldsymbol{\mathcal{M}}$

---

## 4    CHALLENGES WITH PER-STEP LOSS

One intuitive approach to pruning diffusion and flow models is to use a per-step loss. Specifically, one can design a reconstruction loss to have the following form:

$$L = \sum_i \sum_t L_t = \sum_i \sum_t \|f(x_{i,t-1}, \boldsymbol{\mathcal{M}}) - x_{i,t}\|_2^2, \tag{4}$$

where $f$ represents the masked model using mask $\boldsymbol{\mathcal{M}}$, and $x_{i,t}$ is the intermediate output of the $i^{th}$ data sample at step $t$. Notably, this loss closely resembles the standard training loss for diffusion and flow models. However, using a per-step loss presents several challenges. Firstly, it treats losses at all steps as equally important, potentially causing the model to undervalue more critical generation steps. This issue has also been observed in prior diffusion model pruning methods that estimate neural importance. For instance, Fang et al. (2023) introduce re-weighting factors to aggregate influence over steps. Although effective, this approach increases complexity and limits its adaptability to other models. Secondly, per-step loss implicitly assumes correct input at every step, leading to pruning decisions that prioritize short-term accuracy while ignoring the long-term influence of a neuron. This myopic pruning can degrade overall performance. All these limitations can be circumvented with an end-to-end pruning objective.

## 5    OVERVIEW OF ECODIFF

### 5.1    END-TO-END PRUNING OBJECTIVE

We formulate an end-to-end pruning objective that considers the entire denoising process holistically. The pruning objective is to learn a set of masking parameters $\boldsymbol{\mathcal{M}} = [\boldsymbol{\mathcal{M}}_1, ..., \boldsymbol{\mathcal{M}}_N]$ for $N$ target layers to minimize the difference between the final denoised latent $z_0$ generated by the unmasked latent denoising backbone $\epsilon_\theta$ and the predicted $\hat{z}_0$ generated by the masked backbone $\epsilon_\theta^{\text{mask}}$ under the same text prompt $x$ and initial noisy latent $z_T$. To formulate the end-to-end pruning objective, we first define the complete denoising process as $\mathcal{F}$ based on Equation 1:

$$z_0 = f(f(\dots f(z_i, y, 1), y, 2), \dots, y, T) = \mathcal{F}(z_T, y, T) \tag{5}$$

We omit $T$ for the subsequent discussion as it is a constant. The denoising process iteratively refines and denoises the latent representation, starting from the initial time step $T$ and proceeding to $t = 0$, resulting in the final denoised latent $z_0$. Based on Equation 5, we summarize the pruning objective as follows:

$$\arg\min_{\boldsymbol{\mathcal{M}}} \mathbb{E}_{z_T, y \sim \mathcal{C}} \left[ \|\mathcal{F}_{\epsilon_\theta}(z_T, y) - \mathcal{F}_{\epsilon_\theta^{\text{mask}}}(z_T, y, \boldsymbol{\mathcal{M}})\|_2 \right] + \beta\|\boldsymbol{\mathcal{M}}\|_0 \tag{6}$$

where $z_T \sim \mathcal{N}(0, 1)$ is the initial noise, $\mathcal{C} = \{y^i\}_{i=1}^N$ is the dataset consisting of conditions for conditioned generation, $\|\boldsymbol{\mathcal{M}}\|_0 = \sum_{j=1}^{|\boldsymbol{\mathcal{M}}|} \mathbb{I}(\mathcal{M}_j \neq 0)$ denotes as $L_0$ norm for sparsity, and $\beta$ is regularization coefficient of the sparsity regularization. Algorithm 1 shows the procedure of end-to-end pruning using gradient-based optimization.

## 5.2 STRUCTURAL PRUNING VIA NEURON MASKING

Structural pruning has one advantage over nonstructural pruning, as it does not require special hardware support. This work focuses on pruning neurons in transformer blocks. To incorporate sparsity, we apply a learnable discrete pruning mask $\boldsymbol{\mathcal{M}} \in \{0,1\}^n$ on certain neurons of MHA and FFN, which is inspired by LLM-pruner and related methods (Ma et al., 2023; Kwon et al., 2022b; Sun et al., 2023). For MHA, we apply the pruning mask on each attention head as shown in Figure 4a, resulting in a masked multi-head attention (MHA) defined as:

$$\text{MHA}_{\text{mask}}(Q, K, V, \boldsymbol{\mathcal{M}}) = (\mathcal{M}_1 \cdot \text{attn}_1 \| \ldots \| \mathcal{M}_h \cdot \text{attn}_h) W^o, \tag{7}$$

where $\mathcal{M}_i \in \{0,1\}$ controls the output of each head, allowing for head-wise pruning based on learned values. The masked FFN, as shown in Figure 4b, applies a mask $\boldsymbol{\mathcal{M}}_{\text{ffn}} \in \{0,1\}^{d_{\text{ff}}}$ on the neurons after the activation layer, resulting in:

$$\text{FFN}_{\text{mask}}(x, \boldsymbol{\mathcal{M}}_{\text{ffn}}) = (\sigma(xW_1 + b_1) \odot \boldsymbol{\mathcal{M}}_{\text{ffn}})W_2 + b_2, \tag{8}$$

where $\odot$ denotes the Hadamard product. This masking design of MHA and FFN will not change the input and output dimensions of a pruned module, resulting in easier deployment with fewer modifications on a pruned model.

Importantly, once a neuron or attention head is structurally removed from the model parameters, it is absent for all forward passes. Since diffusion and flow models reuse the same denoising network at every timestep, the pruned architecture is naturally shared across the entire sampling trajectory. As a result, the learned mask $\boldsymbol{\mathcal{M}}$ is inherently time-independent, and no timestep-specific masks are required.

## 5.3 CONTINUOUS RELAXATION OF DISCRETE MASKING

Our pruning objective in Equation 6 involves the calculation of $L_0$ norm, which is not differentiable. Hence, we adopt the continuous relaxation of the sparse optimization by applying hard-concrete sampling originally proposed by Louizos et al. (2018). For a scalar continuous masking variable $\hat{\mathcal{M}} \in [0,1]$, the hard-concrete sampling is as follows:

$$s = \sigma\left((\log(u + \delta) - \log(1 - u + \delta) + \lambda)/\alpha\right), \tag{9}$$

$$\hat{\mathcal{M}} = \min(1, \max(0, s(\zeta - \gamma) + \gamma)). \tag{10}$$

The hard-concrete distribution is controlled by the stretch parameters $\gamma$ and $\zeta$, and the temperature parameter $\alpha$. Detailed definitions and specific values for these hyperparameters $(\gamma, \zeta, \alpha)$ are provided in Appendix A, ensuring the reproducibility of our mask learning mechanism. Hence, we can optimize $\boldsymbol{\lambda} \in \mathbb{R}^{|\boldsymbol{\mathcal{M}}|}$ in a differentiable way. The first term in Equation 6 can be formatted as a reconstruction loss $\mathcal{L}_E$:

$$\mathcal{L}_E(\boldsymbol{\lambda}) = \sum_y \sum_{z_T} \|\mathcal{F}_{\epsilon_\theta}(z_T, y) - \mathcal{F}_{\epsilon_\theta^{\text{mask}}}(z_T, y, \hat{\boldsymbol{\mathcal{M}}}(\boldsymbol{\lambda}))\|_2 \tag{11}$$

The $L_0$ complexity loss $\mathcal{L}_0$ given the hard-concrete parameter $\boldsymbol{\lambda}$ can be described as:

$$\mathcal{L}_0(\boldsymbol{\lambda}) = \sum_{j=1}^{|\boldsymbol{\lambda}|} \text{Sigmoid}\left(\lambda_j - \alpha \log \frac{-\gamma}{\zeta}\right), \tag{12}$$

A detailed derivation is in Appendix A. Finally, the end-to-end pruning loss for $\boldsymbol{\lambda}$ is formulated as:

$$\mathcal{L}(\boldsymbol{\lambda}) = \mathcal{L}_E(\boldsymbol{\lambda}) + \beta \mathcal{L}_0(\boldsymbol{\lambda}), \tag{13}$$

After learning the continuous mask control variable $\boldsymbol{\lambda}$, we obtain the final discrete mask $\boldsymbol{\mathcal{M}}$ by applying a threshold $\tau$:

$$\boldsymbol{\mathcal{M}}(\boldsymbol{\lambda}) = \mathbb{I}(\boldsymbol{\lambda} > \tau), \tag{14}$$

where $\tau$ is selected to achieve a desired sparsity ratio. This hard mask physically removes the corresponding neurons, guaranteeing a real structural reduction and commensurate wall-clock speedup at deployment time.

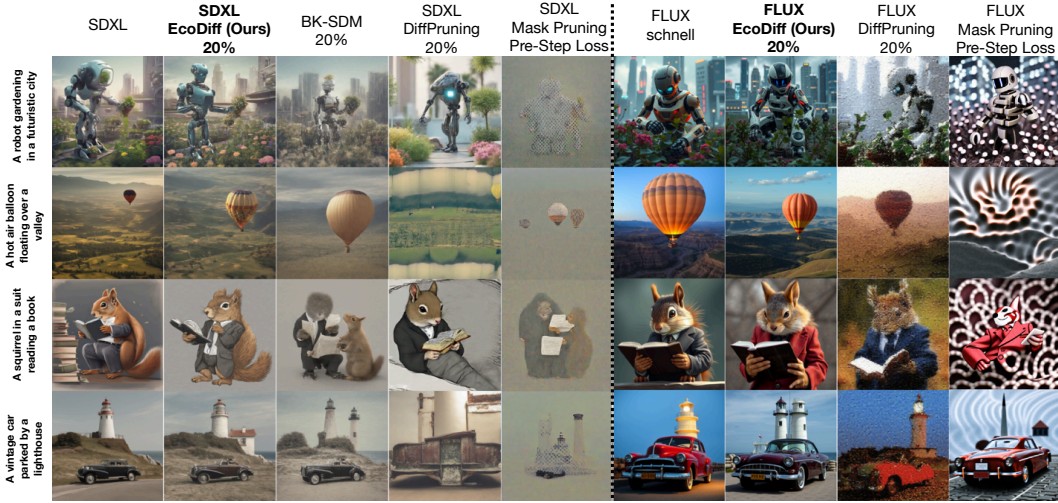

Figure 3: **Example images for comparison.** We show results of pruned SDXL and FLUX-Schnell models. We intentionally use ChatGPT to generate prompts with rich semantics. All methods shown here use the same compute budget of 10 A100 hours.

## 5.4 TIME STEP GRADIENT CHECKPOINTING

To perform the end-to-end pruning shown in Algorithm 1, the backpropagation needs to traverse all generation steps. This necessitates storing all intermediate variables across all steps, leading to a substantial increase in memory usage during mask optimization. We implement time-step gradient checkpointing to address the memory challenge. Traditional gradient checkpointing reduces memory usage by storing selected intermediate outputs and recomputing forward passes between checkpoints (Chen et al., 2016; Gruslys et al., 2016). However, traditional checkpointing only stores intermediate values within a single forward pass, thereby limiting its utility for diffusion models, which require multiple model forward passes across time steps. To address this, we design *time step gradient checkpointing*, which stores intermediate denoised latent $\hat{z}$ after each

---

**Algorithm 2:** Time Step Gradient Checkpointing

**Require:** masked diffusion model $\epsilon_\theta^{\text{mask}}$, target latent $z_0$, diffusion time steps $t = 1, 2, \ldots, T$, masking parameters $\boldsymbol{\lambda}$, loss function $\mathcal{L}$, learning rate $\eta$.

1: $\frac{d\mathcal{L}}{d\boldsymbol{\lambda}} = 0, \mathcal{H} \leftarrow \emptyset$      ▷ Memory initialization
2: **for** $t = T, T-1, \ldots, 1$ **do**
3:     $\hat{z}_{t-1} \leftarrow \epsilon_\theta^{\text{mask}}(z_t, \boldsymbol{\lambda})$     ▷ Calculate latent
4:     $\mathcal{H} \leftarrow \mathcal{H} \cup \{\hat{z}_{t-1}\}$     ▷ Store denoised latent
5: **end for**
6: $\mathcal{L} = \mathcal{L}_E(\hat{z}_0, z_0) + \beta\mathcal{L}_0(\boldsymbol{\lambda})$     ▷ Calculate loss
7: $\frac{d\mathcal{L}}{d\boldsymbol{\lambda}} += \frac{d\mathcal{L}}{d\hat{z}_0}\frac{d\hat{z}_0}{\boldsymbol{\lambda}}$
8: **for** $t = 1, 2, \ldots, T-1$ **do**
9:     $\hat{z}_{t-1} \leftarrow \epsilon_\theta^{\text{mask}}(z_t, \boldsymbol{\lambda})$     ▷ Recompute
10:     $\frac{d\mathcal{L}}{d\hat{z}_t} = \frac{d\mathcal{L}}{d\hat{z}_{t-1}}\frac{d\hat{z}_{t-1}}{d\hat{z}_t}$     ▷ Backprop
11:     $\frac{d\mathcal{L}}{d\boldsymbol{\lambda}} += \frac{d\mathcal{L}}{d\hat{z}_t}\frac{d\hat{z}_t}{d\boldsymbol{\lambda}}$     ▷ Accumulate
12: **end for**
13: **return** $\frac{d\mathcal{L}}{d\boldsymbol{\lambda}}$

---

denoising step. For a diffusion process with $T$ steps, end-to-end backpropagation has a memory complexity of $O(T)$. By implementing time-step gradient checkpointing, we reduce memory complexity to $O(1)$, making it independent of $T$. The runtime complexity remains $O(T)$, with the only added cost being an additional forward pass to recompute activations as needed. We present Algorithm 2 to show how to apply time step gradient checkpointing to calculate the loss gradient w.r.t. the mask control variable $\boldsymbol{\lambda}$.

## 5.5 LIGHT POST-PRUNING RETRAINING

After applying a learned mask to prune the model, we optionally perform a post-pruning retraining stage to recover any residual performance degradation. We explore two complementary strategies: LoRA adaptation, which fine-tunes a small set of low-rank parameters while keeping the model frozen, and full-model fine-tuning, which updates all weights using the standard diffusion training objectives. Both approaches require significantly less computation than training from scratch and offer an efficient way to restore quality when pruning becomes more aggressive. This optional

Table 1: **Quantitative analysis of pruned diffusion models on** $5,000$ **MS COCO and Flickr 30K datasets.** We aim to prune the latest and largest models. Our pruned model achieves a quality comparable to unpruned models at 20% sparsity, demonstrating high semantic fidelity and image quality. All methods use the same compute budget of 10 A100 hours except FLUX-Lite, which uses 1120 H200 hours.

| Models | Methods | Sparsity | #Param | Speedup | MS COCO | | | Flickr 30K | | | Compute Budget |
| | | | | | FID ↓ | CLIP ↑ | SSIM ↑ | FID ↓ | CLIP ↑ | SSIM ↑ | |
|---|---|---|---|---|---|---|---|---|---|---|---|
| **SDXL** Diffusion U-Net | Original | 0% | 2.6B | 1× | 27.43 | 0.33 | 1 | 33.95 | 0.36 | 1 | - |
| | BK-SDM | 20% | 2.1B | 1.25x | 42.87 | 0.30 | 0.47 | 56.17 | 0.32 | 0.46 | 10hr |
| | DiffPruning | 20% | 2.1B | 1.25x | 83.81 | 0.25 | 0.36 | 96.53 | 0.26 | 0.36 | 10hr |
| | Per-Step Loss | 20% | 2.1B | 1.25x | 97.36 | 0.22 | 0.31 | 110.53 | 0.23 | 0.31 | 10hr |
| | **EcoDiff** | 20% | 2.1B | 1.25x | 32.19 | 0.33 | 0.61 | 40.91 | 0.35 | 0.60 | 10hr |
| **FLUX** Flow Matching DiT | Dev | 0% | 11.9B | 1× | 28.47 | 0.34 | 1 | 37.82 | 0.35 | 1 | - |
| | DiffPruning | 20% | 9.6B | 1.25x | 40.84 | 0.33 | 0.31 | 48.02 | 0.33 | 0.31 | 10hr |
| | FLUX-Lite | 33% | 8B | 1.49x | 29.36 | 0.34 | 0.80 | 38.17 | 0.35 | 0.82 | 1120hr |
| | Per-Step Loss | 20% | 9.6B | 1.25x | 50.56 | 0.28 | 0.22 | 59.52 | 0.30 | 0.22 | 10hr |
| | **EcoDiff** | 20% | 9.6B | 1.25x | 30.81 | 0.32 | 0.36 | 42.58 | 0.33 | 0.36 | 10hr |
| | Schnell | 0% | 11.9B | 7× | 30.99 | 0.33 | 1 | 39.70 | 0.35 | 1 | - |
| | DiffPruning | 20% | 9.6B | 8.75x | 42.36 | 0.30 | 0.28 | 54.49 | 0.33 | 0.28 | 10hr |
| | Per-Step Loss | 20% | 9.6B | 8.75x | 52.61 | 0.29 | 0.25 | 66.57 | 0.30 | 0.25 | 10hr |
| | **EcoDiff** | 20% | 9.6B | 8.75x | 31.76 | 0.30 | 0.36 | 43.25 | 0.33 | 0.36 | 10hr |

recovery step complements our pruning framework and further improves the usability of EcoDiff in practical deployment scenarios.

## 6 EXPERIMENTS

### 6.1 SETUP

**Model:** We prune state-of-the-art (SOTA) models: SDXL, FLUX-dev, and FLUX-schnell (Rombach et al., 2022; Podell et al., 2024; Black Forest Labs, 2024). These models cover diffusion and flow-matching, as well as UNet and DiT architecture. **Pruning details**: We only use 100 text prompts from GCC3M (Sharma et al., 2018) for pruning, and a computation budget of 10 A100 GPU hours. **Baselines**: We compare our method with the SOTA diffusion pruning methods, Diff-Pruning (Fang et al., 2023), BK-SDM (Kim et al., 2024), and FLUX-Lite (Daniel Verdú, 2024). For BK-SDM, we only run on SD-XL, since the method is designed for U-Net models. We also include the per-step loss variant for mask learning as a baseline. Per-step loss uses identical settings as EcoDiff, except the masking loss is changed to a per-step reconstruction loss. **Evaluation metrics**: We select Fréchet Inception Distance (FID) (Heusel et al., 2017), CLIP score (Radford et al., 2021), and Structural Similarity Index Measure (SSIM) (Wang et al., 2004). We evaluate under $5,000$ MS COCO and Flickr 30k data samples respectively (Lin et al., 2014; Young et al., 2014). All experiments use a single NVIDIA A100 GPU with 80GB VRAM. More details in Appendix B.

### 6.2 MAIN RESULTS

**SDXL.** SDXL is one of the SOTA diffusion models that employs a U-Net architecture with 2.6B parameters (Podell et al., 2024). We compare the pruning results with BK-SDM, DiffPruning, and our per-step loss variant. From Table 1 and Figure 3, EcoDiff outperforms baseline methods under the same computation budget. We also observe low SSIM scores for all our pruned models compared to conventional pruning methods (Fang et al., 2023), with all values below 0.65. This suggests that EcoDiff prioritizes semantic integrity (high FID/CLIP) over pixel-level mimicry (low SSIM). The low SSIM is attributed to subtle, perceptually acceptable shifts in generated textures or fine-detail placement, which are heavily penalized by pixel-level metrics but do not detract from the overall image quality or semantic adherence to the prompt. As shown in Figure 3, the semantic fidelity and fine-grained quality of our generated images are noticeably superior to those of the baseline methods. We explain more in Section 6.5 and provide results on SD2 in Appendix D.

**FLUX-Dev.** Recently, flow-matching has gained attention due to its superior performance. Currently, the best-performing vision generative model is FLUX-Dev, a flow DiT model with 12B pa-

rameters (Black Forest Labs, 2024). Due to the DiT architecture, BK-SDM is not applicable to FLUX. Hence, we compare with DiffPruning and FLUX-Lite, a community-pruned model leveraging redundancy of middle layers. According to Table 1, both EcoDiff and FLUX-Lite preserve the model performance. However, the FLUX-Lite 8B model was derived from a large-scale, post-hoc retraining effort, requiring 1120 hours on H200 GPUs to achieve 33% sparsity (Hugging Face Community, 2024), we highlight this comparison to demonstrate that EcoDiff achieves comparable quality and structural sparsity (20%) using only 10 A100 GPU hours for the mask learning stage, positioning our method as a vastly more efficient and accessible alternative to resource-intensive retraining campaigns.

**FLUX-Schnell.** FLUX-Schnell is the step-distilled model from FLUX-dev that can generate in 4 steps instead of 28 steps. However, the model is also more difficult to fine-tune due to distillation (Miao et al., 2024; Jia et al., 2024), posing challenges for pruning methods that rely on fine-tuning to recover the model performance. Nevertheless, EcoDiff requires less fine-tuning due to its ability to accurately localize redundancy structures. As shown in Table 1, EcoDiff can prune 20% of FLUX-Schnell with only a 0.77 drop in MS COCO FID and a 3.55 drop in Flickr FID, surpassing other baselines. Consequently, EcoDiff can further prune a distilled model to achieve $8.75\times$ speed up compared to the original FLUX-Dev and reduce 20% parameters compared to FLUX-Dev.

## 6.3 COMPLEXITY ASSESSMENTS

We evaluate the runtime and memory consumption of our differentiable mask learning with SD2, as shown in Figure 15. The results demonstrate consistency with the theorem stated in Section 5.4. By using time-step gradient checkpointing, **we significantly reduce memory usage from $O(T)$ to $O(1)$**, as illustrated in Figure 15a). Additionally, Figure 15b) demonstrates that, despite the significantly reduced memory usage, the runtime complexity remains $O(T)$, with only an $2\times$ increase in runtime. The time-step gradient checkpoint enables end-to-end training on resource-constrained devices, including the largest DiT model, FLUX, with just a single 80 GB GPU. We further provide the carbon footprint analysis of performing pruning using EcoDiff in Appendix I.

## 6.4 POST-PRUNING RETRAINING

We investigate the effectiveness of end-to-end differentiable masking by evaluating the post-pruning retraining stage under various sparsity levels on the SDXL model. As shown in Table 2, the learned EcoDiff mask preserves strong generative quality across 25% to 50% sparsity, and light post-pruning retraining can further enhance performance. Notably, EcoDiff is able to learn an effective pruning mask in only 50 iteration steps, enabling rapid convergence before any retraining is applied. At moderate sparsity levels, LoRA-based retraining provides a lightweight recovery path, partially improving FID and CLIP metrics while requiring substantially less compute compared to full-model fine-tuning. However, as sparsity increases, its limited expressivity becomes insufficient to fully restore generation quality, particularly for fine-grained details and semantic alignment. In such cases, full-model retraining can effectively bridge the remaining performance

Table 2: Sparsity and retraining on SDXL

| Sparsity | Post-Pr. Retrain | MS COCO FID↓ | MS COCO CLIP↑ | Flickr30K FID↓ | Flickr30K CLIP↑ | Iter |
|---|---|---|---|---|---|---|
| 0% | – | 27.43 | 0.33 | 33.95 | 0.36 | – |
| 25% | Full | 31.64 | 0.34 | 41.01 | 0.36 | 10k |
| | LoRA | 32.76 | 0.32 | 39.85 | 0.34 | 10k |
| | No | 34.61 | 0.32 | 42.97 | 0.34 | 50 |
| 30% | Full | 31.70 | 0.34 | 41.22 | 0.36 | 10k |
| | LoRA | 34.43 | 0.32 | 40.73 | 0.34 | 10k |
| | No | 36.86 | 0.31 | 46.06 | 0.33 | 50 |
| 35% | Full | 32.34 | 0.33 | 42.00 | 0.36 | 10k |
| | LoRA | 37.87 | 0.31 | 43.89 | 0.34 | 10k |
| | No | 40.53 | 0.30 | 49.88 | 0.33 | 50 |
| 40% | Full | 33.25 | 0.33 | 46.41 | 0.36 | 10k |
| | LoRA | 43.35 | 0.31 | 49.01 | 0.32 | 10k |
| | No | 43.19 | 0.30 | 49.47 | 0.32 | 50 |
| 50% | Full | 34.87 | 0.33 | 48.95 | 0.36 | 10k |
| | LoRA | 53.89 | 0.28 | 50.78 | 0.30 | 10k |
| | No | 81.76 | 0.26 | 86.38 | 0.29 | 50 |

gap while still remaining computationally practical, requiring only a modest additional cost compared to lightweight adaptation. Even at 50% sparsity, EcoDiff recovers substantial generation quality with just 10,000 post-pruning retraining steps, demonstrating that high-fidelity generation can be

maintained with minimal overhead. This highlights EcoDiff's ability to deliver a compelling trade-off between aggressive model compression and strong generative performance, enabling practical deployment even under significant pruning ratios. More results of light retraining after pruning, as well as detailed experimental configurations, are provided in Appendix F.

## 6.5 ABLATION STUDY

The pruning ratio can be flexibly adjusted by applying different thresholds $\tau$ in Equation 14. Figure 9 demonstrates how the quality of the generated image changes with different pruning ratios (without post-pruning retraining). The quality of generated images can be preserved up to a pruning ratio of $20\%$. At pruning ratios of $10\%$ and $20\%$, for the prompt, *A cat and a dog are playing chess*, the generated image even exhibits enhanced semantic meaning compared to the original image, further validating the decreased FID score in Table 1 and Figure 12. With a higher pruning ratio, model performance starts to degrade. More ablation studies are in Appendix G. We show more results of EcoDiff with time step distillation and feature reuse methods in Appendix E.

## 7 CONCLUSION

In this work, we introduce **EcoDiff**, a structural pruning method for generative models that uses differentiable neuron masking. We design an end-to-end pruning objective to consider the generative ability across all denoising steps and preserve the final denoised latent. To address the high memory demands of end-to-end pruning, we propose time step gradient checkpointing, which reduces memory usage by up to 50 times compared to standard training. Results on the most recent SDXL and FLUX models show that EcoDiff can be applied to UNet and DiT models, is adaptive to diffusion models and flow models, and effectively prunes large models with a reasonable computing budget. Additionally, we show that we can prune on top of time step distilled models, further reducing their latency and deployment requirements. Future work can build on EcoDiff to achieve higher pruning ratios.

### ACKNOWLEDGMENTS

This material is based upon work supported by the Air Force Office of Scientific Research under award number FA2386-24-1-4011, and this research is partially supported by the Singapore Ministry of Education Academic Research Fund Tier 1 (Award No: T1 251RES2509).

### BROADER IMPACTS AND ETHICS STATEMENT

This work proposes EcoDiff, a structural pruning framework for large vision generative models that substantially reduces computational and memory costs while preserving generation quality. By enabling efficient pruning with lightweight post-pruning retraining, EcoDiff lowers the hardware and energy requirements for deploying state-of-the-art text-to-image systems, improving accessibility and reducing environmental impact. As with all generative models, ethical risks such as bias and misuse remain; we therefore emphasize responsible deployment of pruned models in accordance with established ethical guidelines and regulatory standards.

### LLM USAGE STATEMENT

We disclose that we used large language models (LLMs) only in a supportive role for polishing and formatting. All technical contributions, decisions, and claims are the authors' own.

### REPRODUCIBILITY STATEMENT

All source code, scripts, and configuration files required to reproduce our experiments are included in the repository. We make the full, documented repository publicly available on GitHub. A detailed description of the datasets and hyper-parameters can be found in the appendix B.

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

# A DERIVATION OF THE DIFFERENTIABLE $L_0$ REGULARIZATION LOSS

In this section, we provide the detailed derivation of the expected $L_0$ complexity loss $\mathcal{L}_0(\boldsymbol{\lambda})$ presented in Equation 12. We adopt the hard-concrete sampling mechanism proposed by (Louizos et al., 2018).

## A.1 THE HARD-CONCRETE SAMPLING MECHANISM

To enable gradient-based optimization of the discrete mask $\mathcal{M} \in \{0, 1\}$, we employ a continuous relaxation via a scalar masking variable $\hat{\mathcal{M}} \in [0, 1]$. Let $s$ be a continuous random variable sampled from the binary concrete distribution. In our implementation (Equation 9), we utilize the numerically stable form:

$$s = \sigma \left( \frac{\log(u + \delta) - \log(1 - u + \delta) + \lambda_j}{\alpha} \right) \tag{15}$$

where:

- $u \sim \mathcal{U}(0, 1)$ is uniform noise.
- $\lambda_j \in \mathbb{R}$ is the learnable location parameter for the $j$-th element.
- $\alpha \in \mathbb{R}^+$ is the temperature parameter (controlling the approximation to the discrete distribution).
- $\delta$ is a constant used for numerical stability.

For the theoretical derivation of the probability mass, we consider the limit where $\delta \to 0$, simplifying $s$ to:

$$s = \sigma \left( \frac{\log(u) - \log(1 - u) + \lambda_j}{\alpha} \right) \tag{16}$$

The mask $\hat{\mathcal{M}}$ is obtained by stretching $s$ to the interval $(\gamma, \zeta)$ and then rectifying it to the range $[0, 1]$ via a hard-sigmoid function (Equation 10):

$$\hat{\mathcal{M}}_j = \min(1, \max(0, \bar{s}_j)) \tag{17}$$

where $\bar{s}_j = s(\zeta - \gamma) + \gamma$. This rectification allows the mask to take exact values of 0 (fully pruned) or 1 (fully kept) when the stretched variable saturates, while retaining continuous values in the interval $(0, 1)$ to facilitate gradient descent.

## A.2 DERIVING THE $L_0$ LOSS

The $L_0$ norm of the mask corresponds to the number of non-zero elements. Since the mask is stochastic, we minimize the expected $L_0$ norm:

$$\mathbb{E}[\|\hat{\mathcal{M}}\|_0] = \sum_{j=1}^{|\boldsymbol{\lambda}|} P(\hat{\mathcal{M}}_j \neq 0) \tag{18}$$

Based on the rectification in Equation (10), the mask element $\hat{\mathcal{M}}_j$ is non-zero if and only if the stretched variable $\bar{s}_j > 0$. Thus, the probability of a gate being active is:

$$P(\hat{\mathcal{M}}_j \neq 0) = P(\bar{s}_j > 0) = P(s(\zeta - \gamma) + \gamma > 0) \tag{19}$$

Rearranging the inequality for $s$:

$$P\left( s > \frac{-\gamma}{\zeta - \gamma} \right) \tag{20}$$

Substituting the definition of $s$ and recalling that $\sigma(x) = \frac{1}{1 + e^{-x}}$, the inequality becomes:

$$\left( 1 + e^{-\frac{\log u - \log(1-u) + \lambda_j}{\alpha}} \right)^{-1} > \frac{-\gamma}{\zeta - \gamma} \tag{21}$$

Let $L(u) = \log u - \log(1 - u)$ be the log-odds of the uniform variable $u$. $L(u)$ follows a standard Logistic distribution. We solve for the condition on $L(u)$:

$$\frac{L(u) + \lambda_j}{\alpha} > \log\left(\frac{-\gamma}{\zeta - \gamma}\right) - \log\left(1 - \frac{-\gamma}{\zeta - \gamma}\right) \tag{22}$$

Recognizing that $1 - \frac{-\gamma}{\zeta - \gamma} = \frac{\zeta}{\zeta - \gamma}$, the RHS simplifies to $\log(\frac{-\gamma}{\zeta})$. Thus:

$$L(u) > \alpha \log\left(\frac{-\gamma}{\zeta}\right) - \lambda_j \tag{23}$$

Since $L(u)$ follows a standard Logistic distribution, the probability $P(L(u) > X)$ is given by the sigmoid function $\sigma(-X)$. Therefore:

$$\begin{aligned} P(\hat{\mathcal{M}}_j \neq 0) &= \sigma\left(-\left(\alpha \log \frac{-\gamma}{\zeta} - \lambda_j\right)\right) \\ &= \sigma\left(\lambda_j - \alpha \log \frac{-\gamma}{\zeta}\right) \end{aligned} \tag{24}$$

Summing over all elements $j$, we arrive at the same $L_0$ complexity loss used in our objective:

$$\mathcal{L}_0(\boldsymbol{\lambda}) = \sum_{j=1}^{|\boldsymbol{\lambda}|} \text{Sigmoid}\left(\lambda_j - \alpha \log \frac{-\gamma}{\zeta}\right) \tag{25}$$

In our experiments, we adopt the standard hyperparameter settings established in prior work. We set the stretch parameters to $\gamma = -0.1$ and $\zeta = 1.1$ following Louizos et al. (2018); Xia et al. (2024). For the temperature parameter, we use $\alpha = 0.83$ following the configuration in Sheared LLaMA (Xia et al., 2024).

## B  DETAILED EXPERIMENT SETTING AND EVALUATION SETTING

**Model:** we prune SOTA latent diffusion models, SDXL, and FLUX, representing various latent diffusion architectures (Rombach et al., 2022; Podell et al., 2024; Black Forest Labs, 2024). For FLUX, we employ the time-step distillation model, *FLUX.1-schnell*. **Pruning details**: we only use text prompts from GCC3M (Sharma et al., 2018) for training. **Baselines**: we compare our method with the SOTA diffusion pruning methods, DiffPruning (Fang et al., 2023) and BK-SDM (Kim et al., 2024). For BK-SDM, we only run on SD-XL, since the method is constrained for UNet models. To ensure a fair evaluation, we perform the pruning under the same computation budget of 10 A100 GPU hours and 100 data samples. **Evaluation metrics**: we select FID (Heusel et al., 2017), CLIP score (Radford et al., 2021), and SSIM (Wang et al., 2004). We evaluate SSIM by comparing the pruned and original (unmasked) models. We use pretrained CLIP encoder *ViT-B-16* to calculate the CLIP score. For quantitative evaluation, we use the MS COCO and Flickr 30k datasets (Lin et al., 2014; Young et al., 2014), randomly selecting 5,000 image-caption pairs from each. Additionally, we measure computation in Giga Floating Point Operation (GFLOP) and the total number of parameters in the pruned models. **Miscellaneous:** To ensure reproducibility, we use the Hugging Face Diffusers and Accelerator library (Hugging Face, 2021; 2022). All experiments are conducted on a single NVIDIA A100 GPU with 80G VRAM.

By default, we set the masking value $\lambda$ to 5, ensuring that the effective $\mathcal{M}$ is approximately 1 with a high probability to facilitate smoother training. Table 3 and Table 4 are the default sampling training configurations.

### B.1  EVALUATION CONFIGURATION

We conduct extensive experiments to evaluate the performance of **EcoDiff** using both FID and CLIP scores, as summarized in Table 1. For the CLIP score, we use a pretrained **ViT-B/16** model as the backbone. For the FID score, we utilize the FID function from **torchmetrics** with its default settings (e.g., the number of features set to 2048). To accelerate the evaluation process, all images are resized to a uniform resolution of $512 \times 512$ using **bicubic interpolation**.

Table 3: Training Configuration for FLUX

| Parameter | Value |
|---|---|
| Batch size | 4 |
| $lr_{\text{attn}}$ | 0.05 |
| $lr_{\text{ffn}}$ | 1 |
| $lr_{\text{norm}}$ | 0.5 |
| $\beta$ | 0.1 |
| $\delta$ | 0.1 |
| Optimizer | Adam |
| Training Steps | 400 |
| Weight decay | $1 \times 10^{-2}$ |
| Scheduler | constant |
| Diffusion pretrained weight | FLUX.1-schnell |
| Hardware used | $1 \times$ NVIDIA A100 |

Table 4: Training Configuration for SDXL

| Parameter | Value |
|---|---|
| Batch size | 4 |
| $lr_{\text{attn}}$ | 0.15 |
| $lr_{\text{ffn}}$ | 0.15 |
| $lr_{\text{norm}}$ | 0 |
| $\beta$ | 0.5 |
| $\delta$ | 0.5 |
| Optimizer | Adam |
| Training Steps | 400 |
| Weight decay | $1 \times 10^{-2}$ |
| Scheduler | constant |
| Diffusion pretrained weight | stable-diffusion-xl-base-1.0 |
| Hardware used | $1 \times$ NVIDIA A100 |

## C  NEURON MASKING

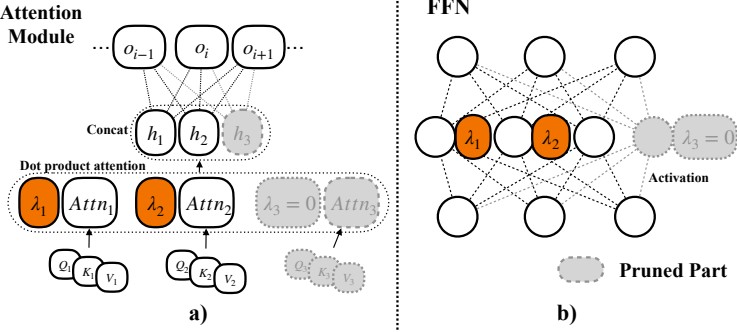

Figure 4: **Neuron masking on MHA and FFN in transformer blocks.** Here, a) illustrates masking within an attention module, and b) shows masking within an FFN. $\lambda$ is the mask variable.

## D  RESULTS ON STABLE DIFFUSION 2

EcoDiff is designed to be versatile, adaptable and model-agnostic. For Stable Diffusion 2 (SD2), we specifically focus our pruning approach on the attention (**Attn**) and feed-forward network (**FFN**)

Table 5: Ablation study of SD2 models. Metrics include FID and CLIP scores on MS COCO and Flickr30K datasets. * denotes as the model pruning ratio on the **Attn** and **FFN** only.

| Model | MS COCO | | Flickr30K | |
|---|---|---|---|---|
| | FID | CLIP | FID | CLIP |
| SD2 | 30.15 | 0.33 | 36.10 | 0.35 |
| EcoDiff Pruning 10%* | 30.61 | 0.32 | 38.12 | 0.34 |
| EcoDiff Pruning 20%* | 31.16 | 0.32 | 42.67 | 0.34 |

Table 6: Ablation study of Flux models, *Flux-dev*, *Flux-schnell*. Metrics include FID and CLIP scores on MS COCO and Flickr30K datasets.

| Model | MS COCO | | Flickr30K | |
|---|---|---|---|---|
| | FID | CLIP | FID | CLIP |
| FLUX Dev | 35.59 | 0.32 | 45.68 | 0.34 |
| FLUX-Schnell Step-Distilled | 30.99 | 0.33 | 39.70 | 0.35 |
| FLUX EcoDiff Pruning 10% | 32.16 | 0.32 | 42.56 | 0.34 |
| FLUX EcoDiff Pruning 15% | 31.76 | 0.30 | 43.25 | 0.33 |

blocks. Table 5 highlights the model-agnostic feature of EcoDiff, demonstrating its ability to optimize diverse architectures without requiring model-specific adjustments or design changes.

# E COMPATIBILITY WITH OTHER METHODS

## E.1 COMPATIBILITY WITH STEP DISTILLATION

Figure 5 presents the comparison results between *FLUX-Dev*, FLUX with timestep distillation (*FLUX-schnell*), *FLUX-schnell* with 10% pruning using EcoDiff, and *FLUX-schnell* with 15% pruning using EcoDiff. The results demonstrate that EcoDiff is compatible with the timestep-distilled model, effectively preserving semantic fidelity while maintaining fine-grained detail. Notably, we occasionally observed that the quality of images generated by the pruned model with EcoDiff sometimes outperformed the timestep-distilled model (with prompt, *A floating city above the clouds at sunset.*), producing results that were semantically more aligned with the *FLUX-Dev* model, which requires 50 intervention steps.

## E.2 COMPATIBILITY WITH FEATURE REUSE

EcoDiff demonstrates high versatility by efficiently **integrating with feature reuse methods like DeepCache** (Ma et al., 2024b) as shown in Figure 6. Additionally, combining EcoDiff with Deep-Cache enables significant reductions in effective **GFLOPs** and runtime, highlighting its compatibility and effectiveness in optimizing resource usage.

Table 7: Training Configuration for post-pruning retraining. The left table shows the configuration for LoRA retraining. The right table shows the configuration for Full-model retraining.

(a) Training Configuration for LoRA retrain

| Parameter | Value |
|---|---|
| Batch size | 32 |
| Resolution | 512 |
| Rank | 16 |
| $lr$ | $1 \times 10^{-6}$ |
| Optimizer | Adam |
| Dataset | 100 prompts synthesized from GCC3M |
| Training Steps | 10000 |
| Weight decay | $1 \times 10^{-2}$ |
| Scheduler | constant |
| Hardware used | $1 \times$ NVIDIA A100 |

(b) Training Configuration for Full-model retrain

| Parameter | Value |
|---|---|
| Batch size | 32 |
| Resolution | 1024 |
| $lr$ | $1 \times 10^{-7}$ |
| Optimizer | Adam |
| Dataset | 10000 prompts synthesized from GCC3M |
| Training Steps | 10000 |
| Weight decay | $1 \times 10^{-2}$ |
| Scheduler | constant |
| Hardware used | $1 \times$ NVIDIA A100 |

# F  MORE RESULTS ON POST-PRUNING RETRAINING

## F.1  DETAILS OF POST-PRUNING ADAPTATION

We provide additional details and results on the post-pruning retraining experiments described in Section 6.4. Figure 7, 8 and Table 6 illustrate that the pruned model's performance with EcoDiff begins to degrade beyond a $20\%$ pruning ratio. While semantic fidelity is largely preserved, fine-grained visual details deteriorate, as illustrated by generations from prompts like *A robot dog exploring an abandoned spaceship*, *A mystical wolf howling under a glowing aurora*, and *A cozy library with a roaring fireplace*.

To mitigate this degradation, we adopt two complementary retraining strategies: LoRA adaptation and full-model fine-tuning. In both cases, retraining is performed for 10,000 steps and requires approximately 10 hours of additional compute. Importantly, the training images are not sourced from the original dataset but are instead synthesized by the original SDXL model using prompts from GCC3M (Sharma et al., 2018).

For LoRA fine-tuning, we train with 100 prompts from GCC3M and fine-tune a small number of low-rank parameters. The complete training configuration is provided in Table 7a. As shown in Figure 7 and Table 2, this lightweight approach provides moderate improvements in FID and CLIP scores, recovering a significant portion of the lost semantic fidelity and restoring many fine-grained details with minimal computational overhead. However, as pruning ratios increase, the limited expressivity of LoRA constrains its ability to fully recover generation quality, particularly for complex structural or textural details.

For full-model fine-tuning, we scale up the retraining process using 10,000 text prompts from GCC3M, with the corresponding training images synthesized by the original SDXL model. All model parameters are updated with the standard diffusion training objective. Table 7b summarizes the full retraining configuration details. As illustrated in Figure 8, this approach further enhances generation quality, particularly at higher sparsity levels ($40$–$50\%$), and brings the pruned model's performance closer to that of the original.

Overall, these results confirm that EcoDiff's pruning framework is highly compatible with practical retraining pipelines. Even with a modest amount of additional compute, both LoRA and full-model fine-tuning significantly narrow the quality gap introduced by pruning, enabling aggressive compression while preserving both semantic fidelity and visual detail.

## F.2  POST-PRUNING RETRAINING ABLATION ON DATASETS

We further validate our post-pruning retraining strategy by investigating the impact of the retraining dataset on performance recovery. In the main experiments, we utilized images synthesized from the GCC3M dataset to serve as a practical, low-cost proxy for the original, large-scale training distribution.

Table 8: **Ablation study on $\beta$.** Evaluated on Flickr30K dataset with a fixed pruning ratio of 15% on FLUX.

| $\beta$ | 0.001 | 0.005 | 0.01 | 0.1 | 0.5 |
|---|---|---|---|---|---|
| **FID** ↓ | 43.21 | 44.50 | 43.24 | 87.5 | 323.1 |
| **CLIP** ↑ | 0.32 | 0.34 | 0.33 | 0.21 | 0.03 |

Table 9: **Ablation study on $\beta$.** Evaluated on MS COCO dataset with a fixed pruning ratio of 20% on SDXL

| $\beta$ | 0.01 | 0.1 | 0.5 | 1 | 2 |
|---|---|---|---|---|---|
| **FID** ↓ | 41.23 | 34.23 | **33.74** | 67.78 | 121.2 |
| **CLIP** ↑ | 0.29 | 0.30 | **0.30** | 0.221 | 0.19 |

To address potential concerns that this approach might limit the validity of our recovery claims, we conducted an additional ablation study using a public subset of the LAION2B-en-aesthetic dataset (Schuhmann et al., 2022), derived from the LAION-5B corpus. This subset is generally considered to share a distribution closer to the models' original training data than synthetic imagery.

The results, presented in Table 10, demonstrate that retraining on the Laion subset yields performance recovery consistent with our initial findings on synthetic data. Specifically, at 50% sparsity, full-model retraining reduces the MS COCO FID to 32.86 and the Flickr30K FID to 35.69. This consistency strengthens our claim that the post-pruning adaptation whether using synthesized prompts or a relevant public dataset subset is an effective, practical strategy for recovering generative quality, and our core claims are not an artifact of the synthetic data generation pipeline.

# G    MORE ABLATION STUDY

## G.1    DEMO IMAGES ON ABLATION STUDY

## G.2    ABLATION STUDY ON REGULARIZATION COEFFICIENT ($\beta$)

We conducted ablation studies on the regularization coefficient $\beta$ (Equation 13), which is essential for balancing semantic fidelity (reconstruction loss $\mathcal{L}_E$) against model sparsity ($\mathcal{L}_0$ complexity loss). Performance is highly sensitive to the magnitude of $\beta$, confirming the need for careful tuning to achieve an optimal capacity-sparsity trade-off. For SDXL pruned at 20% sparsity (Table 9), the optimal $\beta$ range was found between 0.1 and 0.5. Values outside this range led to significant degradation: lower values failed to enforce sufficient pruning pressure, while excessively high values over-penalized the sparsity term, resulting in severe quality collapse. The study on the FLUX model, tested at 15% sparsity (Table 8), showed a similar pattern but required a significantly smaller optimal $\beta$ range, specifically around 0.005 to 0.01. These results confirm that the optimal $\beta$ value is architecture-dependent and its careful selection is critical for localizing redundancy accurately without causing catastrophic performance degradation.

## G.3    ABLATION ON PRUNING SINGLE MODULES

Figure 11 demonstrates the flexibility of our pruning mask, allowing for adjustable thresholding across different blocks, such as pruning exclusively in the **Attn** or **FFN** blocks. Notably, image quality retains high fidelity regardless of the targeted pruning module. However, minor variations in fine-grained details and semantic meaning are observed, and we will investigate them further in future work.

## G.4    ABLATION ON PRUNING SETTING

We apply a threshold $\tau$ in Equation 14 to We apply a threshold $\tau$ in Equation 14 to mask the head and FFN layers. Thresholding can be applied in two ways: **globally** across all blocks or **locally**

Table 10: Additional results on Sparsity and retraining on SDXL using subset of `laion2B-en-aesthetic` dataset.

| Sparsity | Post-Pr. Retrain | MS COCO | | Flickr30K | | Iter |
|---|---|---|---|---|---|---|
| | | FID↓ | CLIP↑ | FID↓ | CLIP↑ | |
| 0% | – | 27.43 | 0.33 | 33.95 | 0.36 | – |
| 30% | Full | 28.66 | 0.33 | 35.78 | 0.35 | 10k |
| | LoRA | 31.05 | 0.32 | 38.47 | 0.34 | 10k |
| | No | 36.86 | 0.32 | 46.06 | 0.34 | 50 |
| 40% | Full | 30.16 | 0.33 | 35.89 | 0.35 | 10k |
| | LoRA | 45.03 | 0.30 | 50.53 | 0.32 | 10k |
| | No | 43.19 | 0.30 | 49.47 | 0.33 | 50 |
| 50% | Full | 32.86 | 0.32 | 35.69 | 0.34 | 10k |
| | LoRA | 50.38 | 0.28 | 49.55 | 0.30 | 10k |
| | No | 81.76 | 0.26 | 86.38 | 0.29 | 50 |

Table 11: **Ablation study on calibration data size.** Evaluated on MS COCO dataset with a fixed pruning ratio of 15% on FLUX. Only applied mask learning.

| Size | 1 | 8 | 100 | 256 | 512 |
|---|---|---|---|---|---|
| FID ↓ | 29.79 | 31.51 | 31.76 | 77.48 | 33.25 |
| CLIP ↑ | 0.31 | 0.30 | 0.30 | 0.26 | 0.33 |

within each block (e.g., FFN, Norm, Attn). Global thresholding uses a single threshold $\tau$ applied to the entire mask $\mathcal{M}$. In contrast, local thresholding applies $\tau$ individually within each block, treating them independently. To evaluate these approaches, we conduct an ablation study, and the results are shown in Figure 10. The global thresholding approach demonstrates superior results, accounting for the interactions and trade-offs between layers and blocks, leading to more effective masking. In contrast, local thresholding results in a deterioration in quality.

## G.5 ABLATION ON CALIBRATION DATA

We conduct experiments across various training dataset sizes, ranging from 1 to 512, as shown in Table 12. Notably, even when training with a single randomly selected text prompt, the performance degradation relative to a dataset size of 100 remains minimal, suggesting that an appropriate mask of the network can be learned with limited data.

## G.6 ABLATION STUDY ON FLUX

Table 8 and Table 11 present the results of our ablation studies on FLUX. Similar results are observed with the SDXL model. Notably, even with just a single training data point, FLUX can produce relatively competitive results, highlighting its robustness and efficiency in resource-constrained scenarios.

Table 12: Calibration data size ablation.

| Size | 1 | 8 | 64 | 100 | 256 | 512 |
|---|---|---|---|---|---|---|
| FID ↓ | 35.18 | 36.17 | 34.13 | **33.74** | 33.76 | 33.68 |
| CLIP ↑ | 0.30 | 0.31 | 0.31 | 0.30 | 0.31 | **0.31** |

Table 13: Ablation study of SDXL models with different pruning setting.

| Model | MS COCO | | Flickr30K | |
|---|---|---|---|---|
| | FID | CLIP | FID | CLIP |
| SDXL Original | 35.50 | 0.31 | 49.34 | 0.34 |
| EcoDiff 20% Global | 34.41 | 0.31 | 42.84 | 0.33 |
| EcoDiff 20% Local | 92.84 | 0.26 | 117.18 | 0.27 |

Table 14: Carbon Footprint and Running hours for differentiable mask learning.

| Model | Training Time(hours) | VRAM Usage | Carbon Footprint |
|---|---|---|---|
| SD2 | 3 | 4.6G | 12.0g |
| SDXL | 4.4 | 22.9G | 93.0g |
| FLUX | 0.54 | 64.2G | 31.50g |

## H    SEMANTIC CHANGING WITH DIFFERENT PRUNING RATIOS

Figure 14 and Figure 13 show the superior performance of EcoDiff with up to 20% of pruning ratio for SDXL and 15% for FLUX.

## I    CARBON FOOTPRINT ANALYSIS

We conduct a carbon footprint analysis for EcoDiff training. The analysis uses training configurations with only 200 training steps, as most configurations converge within 400 steps. The carbon footprint calculations are based on the default configurations outlined in Appendix B. All calculations follow the methodology used in the SD2 carbon footprint analysis (Rombach et al., 2022).

## J    COMPLEXITY ASSESSMENTS

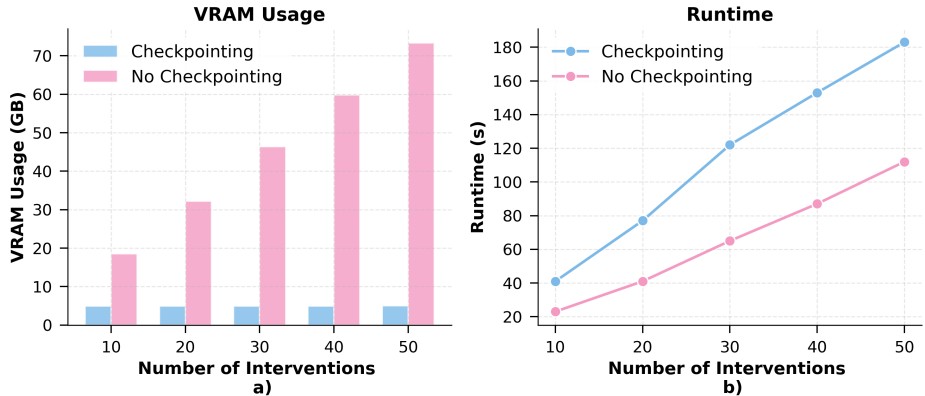

Figure 15: **VRAM usage and runtime per training step comparison with and without gradient checkpointing on SD2.** The measurements are averaged over five runs.

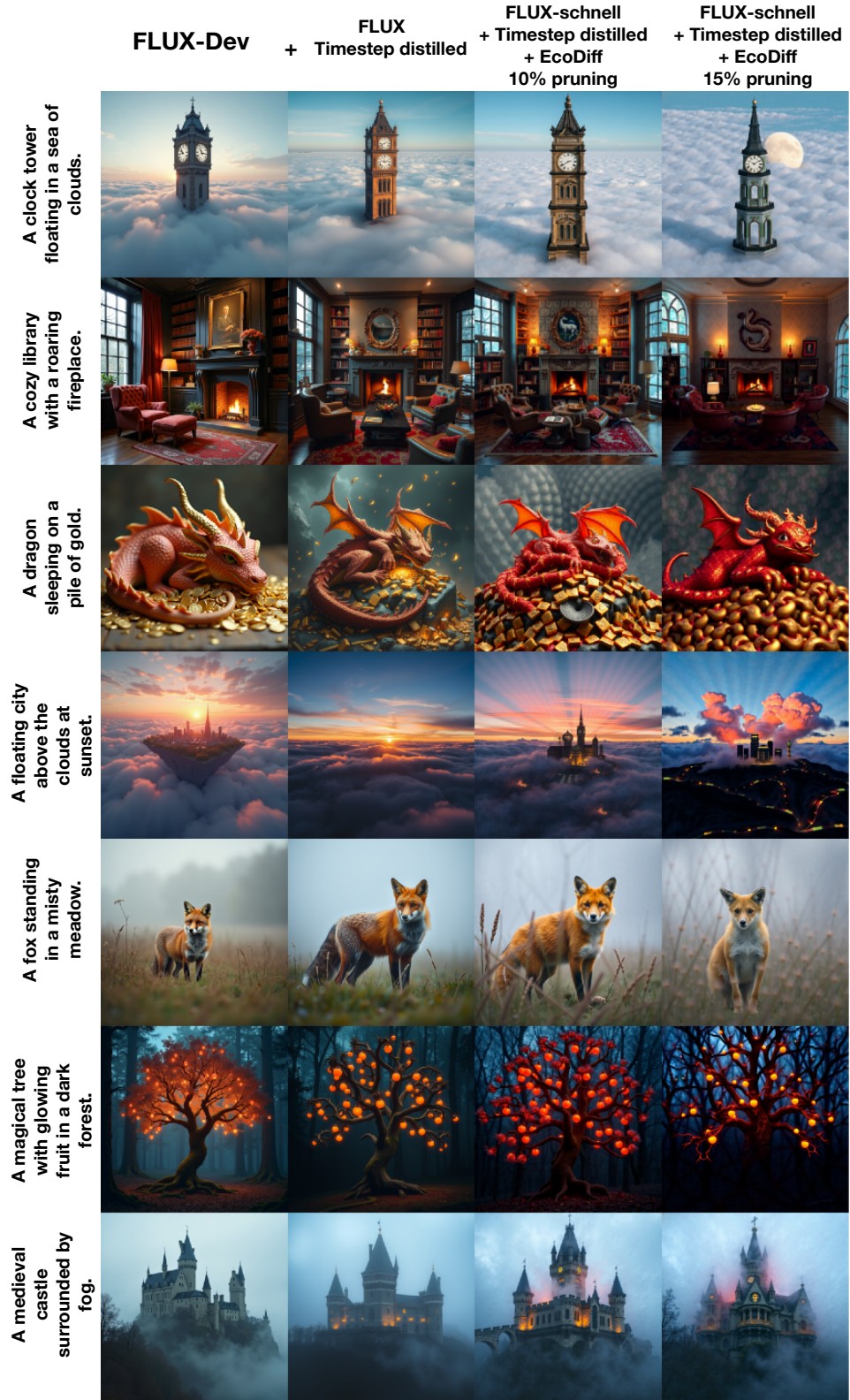

Figure 5: **EcoDiff with timestep-distilled model, FLUX**

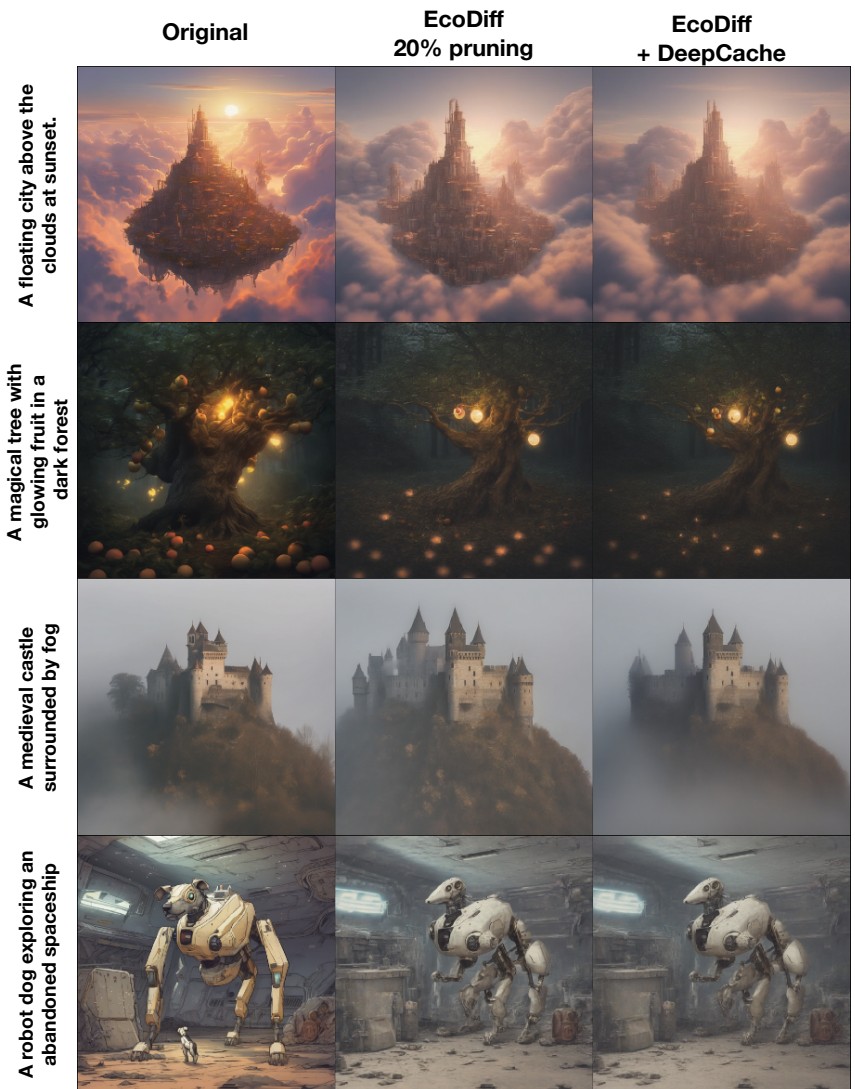

Figure 6: **EcoDiff with DeepCache**

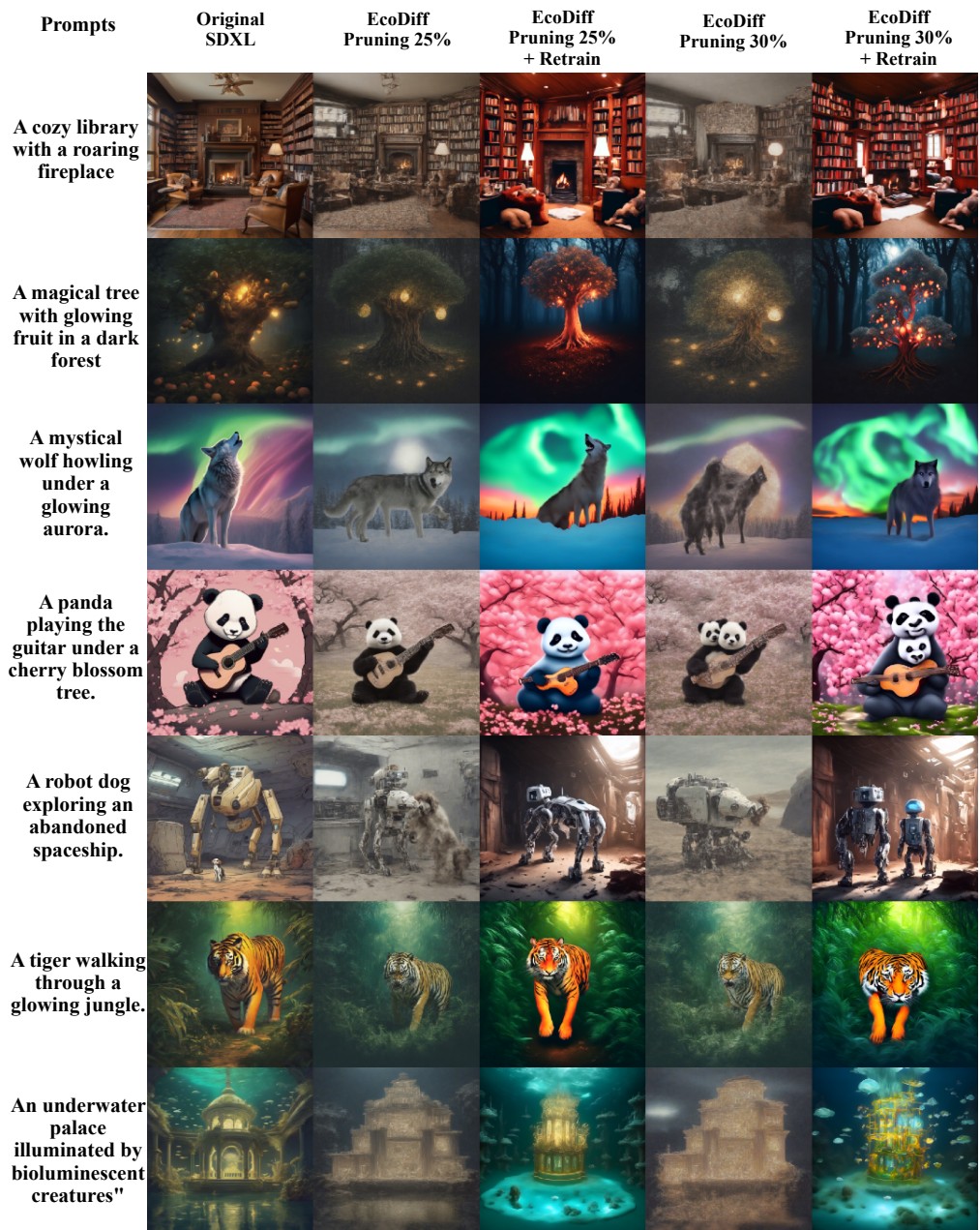

Figure 7: **EcoDiff Pruning with LoRA retraining.**

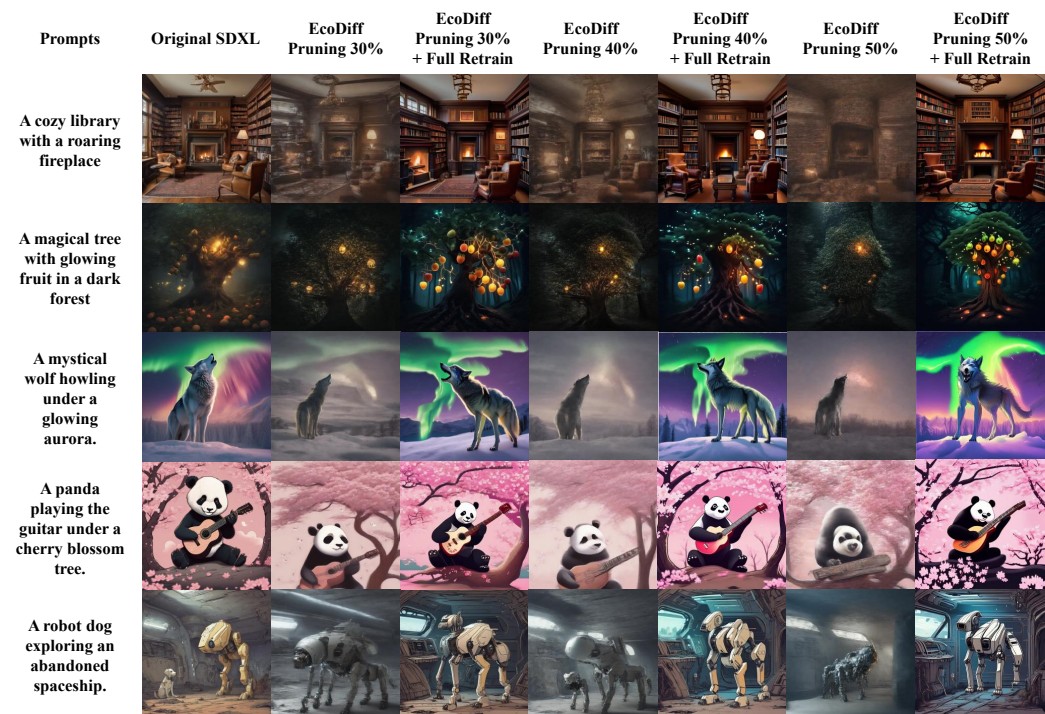

Figure 8: **EcoDiff Pruning with Full-model retraining.**

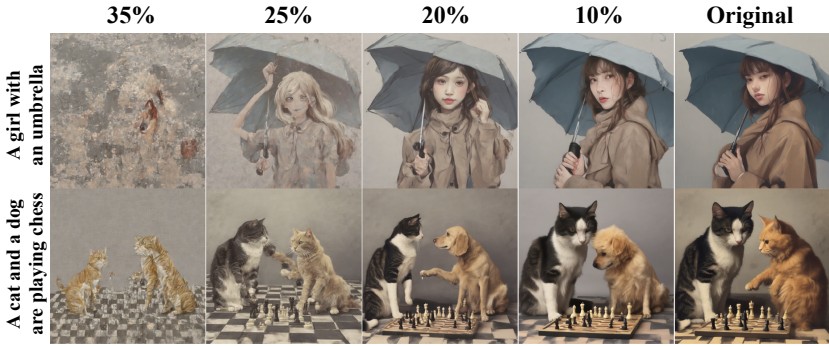

Figure 9: **Mask sparsity ablation without retraining.** Applying a learned mask without retraining shows that EcoDiff accurately identifies redundancy.

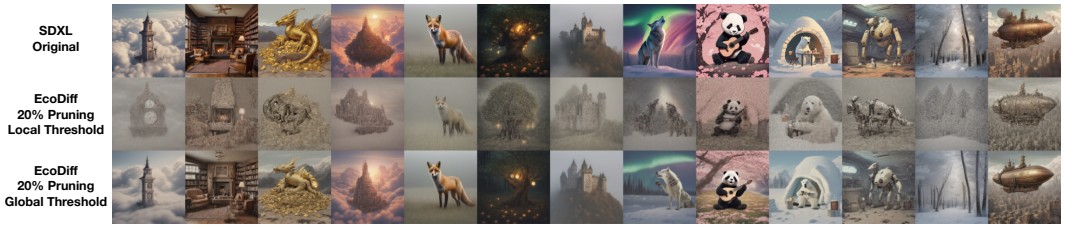

Figure 10: **EcoDiff Pruning with global thresholding and local thresholding**

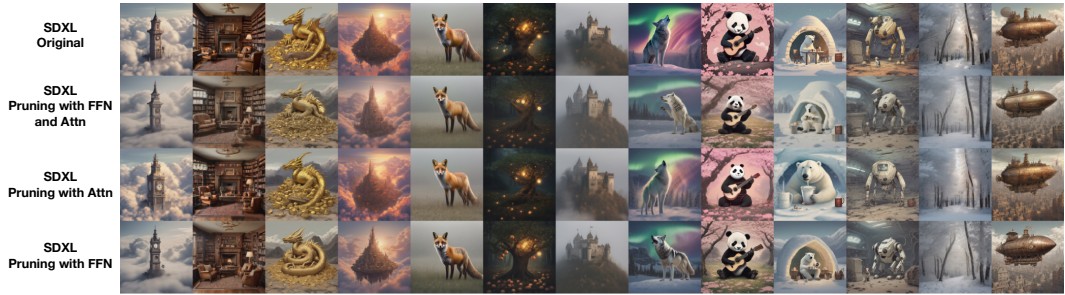

Figure 11: **EcoDiff Prunig with masking on single Module, FFN module or Attn module.**

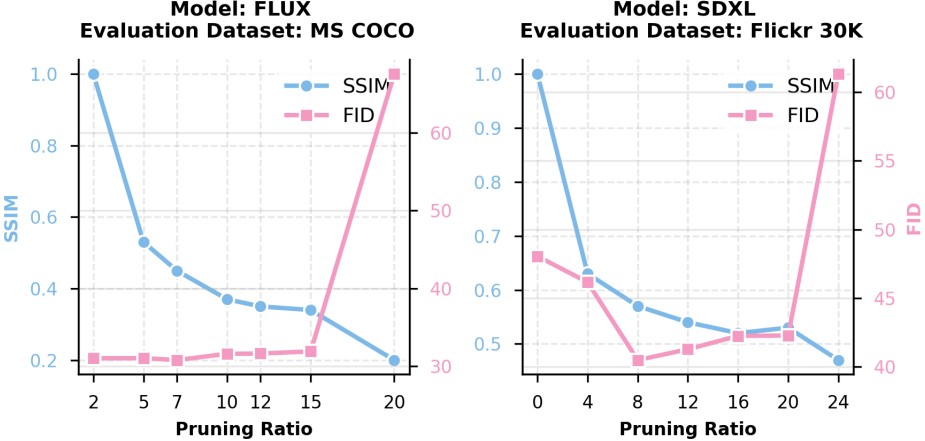

Figure 12: **Evaluation on different pruning ratios**. Models start to degrade at $20\%$ pruning for FLUX and $25\%$ for SDXL.

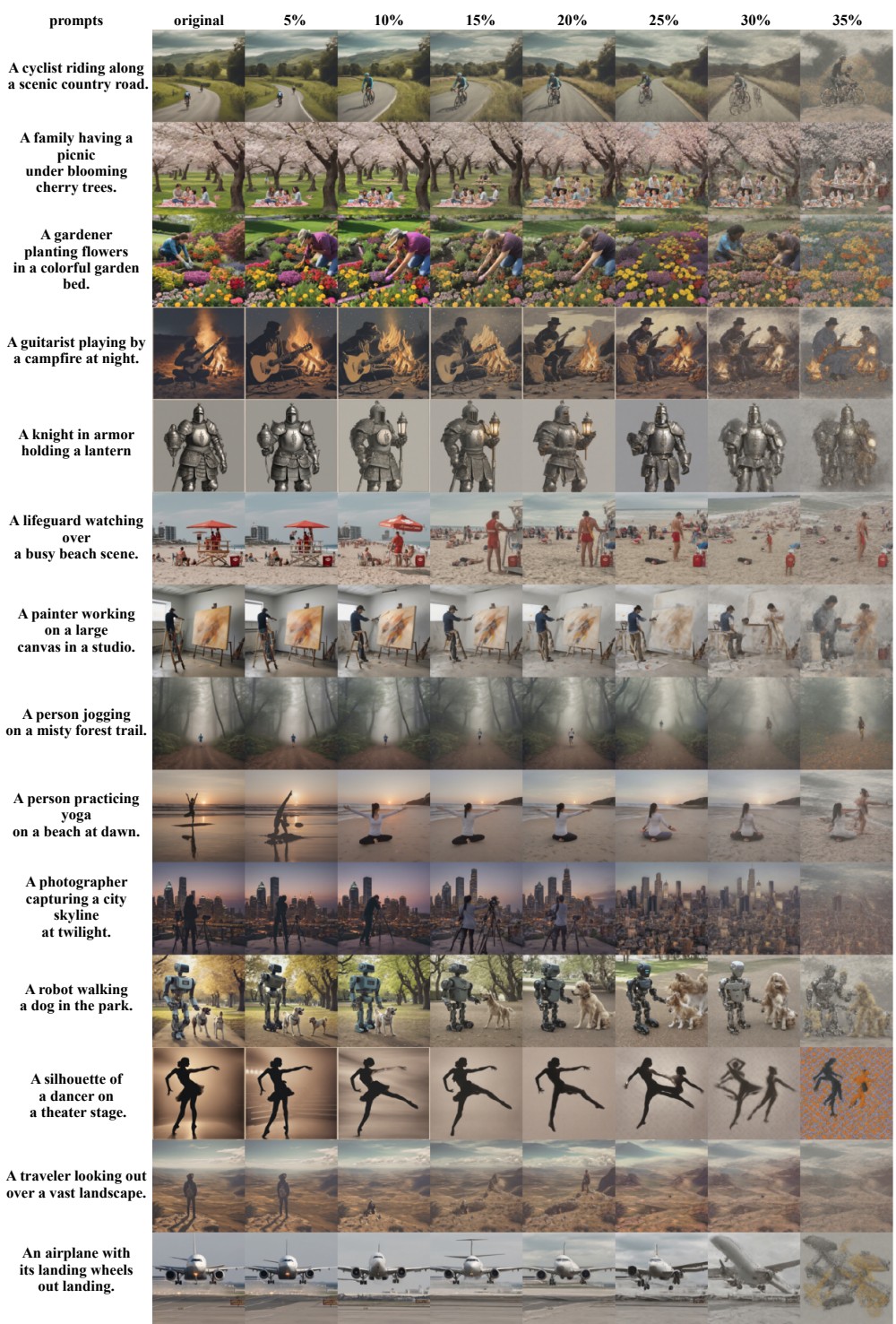

Figure 13: **More Samples with masking diffusion model, *SDXL-base*, with different pruning ratios. No post-prune retraining applied.**

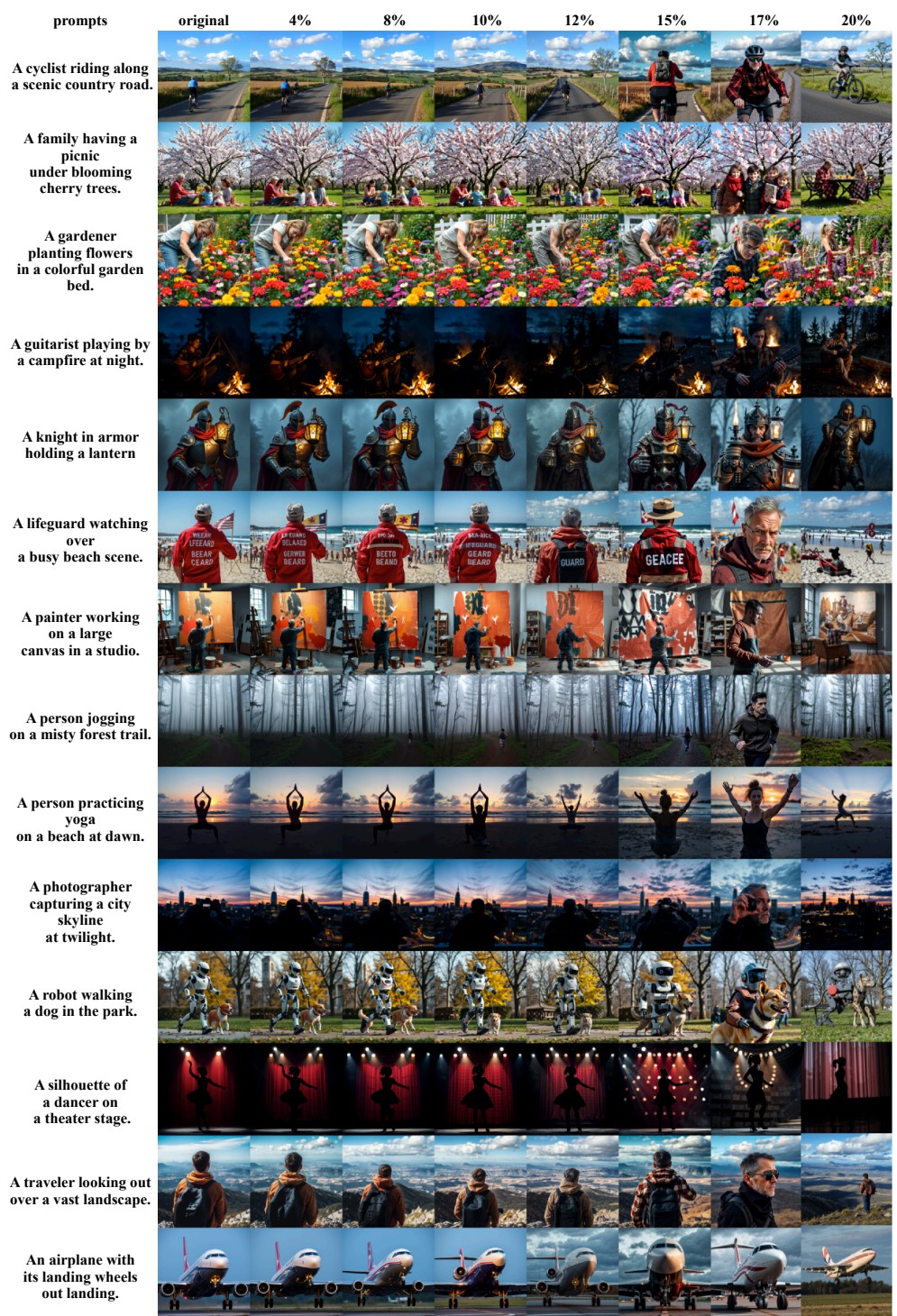

Figure 14: **More Samples with masking diffusion model, *FLUX-schnell*, with different pruning ratios. No post-prune retraining applied.**

