# OpenReview forum: "Learnable Sparsity for Vision Generative Models"
_ICLR.cc/2026/Conference — ICLR 2026 Poster_

### Official Review · Reviewer_h5Na · 2025-10-21

**Soundness:** 2
**Presentation:** 2
**Contribution:** 2
**Rating:** 4
**Confidence:** 3

**Summary:**

This paper introduces **EcoDiff**, an efficient and model-agnostic framework for pruning large vision generative models. Its core innovation is an **end-to-end differentiable masking** scheme that learns to remove redundant parameters by optimizing for final output quality.

**Strengths:**

1. **High Efficiency:** The method achieves up to 20% sparsity on billion-parameter models (SDXL, FLUX) using only **100 samples** and **10 A100 GPU hours**, which is highly impressive given the scale.
2. **End-to-End Pruning Objective:** The proposed holistic loss jointly optimizes all denoising steps, outperforming traditional per-step pruning methods.
3. **Practicality:** The design is model-agnostic and potentially applicable to both diffusion and flow-based generative models.

**Weaknesses:**

### **Theoretical Concerns**

There is a fundamental error in Equation (12). Contrary to what is presented, the correct form of the L₀ regularization term, as established in *Learning Sparse Neural Networks through L₀ Regularization* (Louizos et al., 2018), should be:
$$
L_0(\lambda)=\Sigma \text{Sigmoid}\left(\lambda_i-\alpha\log\frac{-\gamma}{\zeta}\right),
$$
this error propagates to the derivation in Appendix A, which is consequently invalid. The intended relationship should instead be:
$$
L_0(\lambda)=\Sigma \text{Sigmoid}\left(\lambda_i-\alpha\log\frac{-\gamma}{\zeta}\right)\approx \|e^{\lambda}\|_{L^1}.
$$
Moreover, the claim that ||e^lambda|| can be bounded by ||\lambda|| is mathematically unsound. There is no generic upper bound of ||e^lambda|| in terms of ||\lambda||, since the former grows exponentially while the latter only grows linearly.

Furthermore, the motivation for using this inaccurate approximation is not justified. It is both clearer and more rigorous to simply use the exact expression:
$$
L_0(\lambda)=\Sigma \text{Sigmoid}\left(\lambda_i-\alpha\log\frac{-\gamma}{\zeta}\right),
$$
which is well-founded in the literature and avoids unnecessary ambiguity.

### Comment on experiments

**Experimental Fairness**

The comparison with FLUX-Lite is problematic in terms of computational fairness: FLUX-Lite used 1,120 H200 GPU hours, while EcoDiff required only 10 A100 hours. Although this highlights efficiency, it remains unclear whether FLUX-Lite could achieve similar performance under a comparable computational budget.

**Ablation studies on loss parameters are missing.**

The paper lacks ablation studies investigating the impact of key parameters in the proposed loss function (Eq. 13), such as the regularization coefficient `β` and the hard concrete parameters (e.g., δ, γ, ζ). The influence of these choices on the trade-off between sparsity and performance is not analyzed.

### Reading difficulty

The specific implementation of the learnable mask—such as the hard discrete sampling procedure and the selection of parameters (e.g., δ, γ, ζ)—is only briefly described in the main text. Readers must refer to the appendix to fully understand these mechanisms.

**Questions:**

See the Weaknesses section.

---

> ### Author Response · Authors · 2025-11-22
> **Response to Reviewer h5Na (Part 1/1)**
>
> We thank you for your rigorous review and appreciate your acknowledgment of EcoDiff's high efficiency, the strength of our end-to-end pruning objective, and the practicality of our model-agnostic design. Your feedback on the theoretical formulation has been invaluable, and we have taken your concerns very seriously.
>
>
> **Weaknesses:**
>
> > **W1:** Theoretical Concerns. There is a fundamental error in Equation (12)... this error propagates to the derivation in Appendix A, which is consequently invalid…
>
>
> We thank you for the rigorous check and for identifying the typo in Equation 12, which regrettably led to an inaccurate derivation in the original Appendix A. We acknowledge this error and have thoroughly corrected it in the revised manuscript. We confirm and emphasize that our actual implementation correctly uses the exact hard concrete $\mathcal{L}_0$ relaxation loss, as originally proposed by Louizos et al. (2018), which is reflected in Algorithm 2, Line 6.
>
> The error in the manuscript's Eq. (12) and Appendix A occurred only during an unnecessary attempt to derive a supplementary property of the hard concrete relaxation. Crucially, this error did not affect our core method, implementation, or experimental results.
>
> We agree that using the exact, well-founded $\mathcal{L}_0$ relaxation expression (shown below) is both clearer and more rigorous, and we have removed all the inaccurate discussion involving the $\mathcal{L}_1$ norm.
> $$
> L_0(\lambda) = \sum \text{Sigmoid}\left(\lambda_i - \alpha\log\frac{-\gamma}{\zeta}\right)
> $$
>
> Furthermore, we have revised Appendix A to include a proper derivation detailing the relationship between the exact discrete $\mathcal{L}_0$ loss and the hard concrete relaxation used, ensuring theoretical consistency with the established literature (Louizos et al., 2018).
>
>
>
> > **W2:** Experimental Fairness. The comparison with FLUX-Lite is problematic...
>
> We thank you for this point and agree the compute is not normalized. We clarify that the FLUX-Lite 8B model we compared against was trained by a community member (Freepik) who publicly disclosed that they extensively retrained the model for 1120 hours on H200 GPUs (https://huggingface.co/Freepik/flux.1-lite-8B-alpha/discussions/16).
> Our intention was to benchmark against this specific model to highlight that EcoDiff can achieve comparable sparsity (20%) with a fraction of the cost (10 A100 hours). We present this as a practical trade-off: our method offers a highly efficient alternative for scenarios where massive retraining is not feasible.
>
> > **W3:** Ablation studies on loss parameters are missing.
>
> We appreciate you pointing this out. We clarify that ablations for the regularization coefficient $\beta$ were included in the original appendix (Tables 8 and 9). We acknowledge the need for explicit analysis and will add a new section in Appendix to discuss these results.
>
> > **W4:** Reading difficulty. The specific implementation of the learnable mask... is only briefly described...
>
> We agree that the specific implementation of the hard concrete distribution involves several parameters ($\delta, \gamma, \zeta$) and requires clarity. Since this mechanism is well-established in the literature (Louizos et al., 2018), we adopted standard parameter settings to focus on our novel contributions (end-to-end pruning, time-step checkpointing) in the main text. To address your concern without overwhelming the main text, we will add a clearer reference in Section 5.3 to guide readers to the appendix for these details. Specifically, we confirm that we utilized standard, well-vetted values from the literature: we set $\gamma = -0.1$ and $\zeta = 1.1$, (following Louizos et al., 2018 and Xia et al., 2024), and we adopted $\alpha=0.83$ from the Sheared LLaMA work (Xia et al., 2024). Documenting these precise settings will eliminate any ambiguity regarding the mask implementation.
>
> We are sincerely grateful for your thoughtful and constructive feedback, which has helped us further refine and strengthen our paper. We hope that our experiments and revised derivation address your concerns and support a positive re-evaluation of our work.
>
>
>
> References
>
>
> Louizos, C., Welling, M., & Kingma, D. P. (2018). Learning Sparse Neural Networks through $L_0$ Regularization. In International Conference on Learning Representations.
>
>
> Xia, M., Gao, T., Zeng, Z., & Chen, D. (2024). Sheared LLaMA: Accelerating Language Model Pre-training via Structured Pruning. In International Conference on Learning Representations.

---

> > ### Comment · Reviewer_h5Na · 2025-11-27
> >
> > Thank the author for their response. My confusion has been resolved, so I increase my score.

---

> > > ### Author Response · Authors · 2025-11-27
> > >
> > > Thank you very much for your follow-up and for revisiting your assessment. We sincerely appreciate the time and care you dedicated to reviewing our revisions and clarifications. Your feedback has helped us strengthen both the clarity and rigour of the manuscript.
> > >
> > > Thank you again for your thoughtful review and for your updated evaluation.

---

### Official Review · Reviewer_2dT4 · 2025-11-04

**Soundness:** 3
**Presentation:** 3
**Contribution:** 3
**Rating:** 6
**Confidence:** 4

**Summary:**

This paper explores pruning of visual generative models, including LDM and Flux. Existing pruning strategies are coarse-grained and fail to balance performance and sparsity. Therefore, in this paper, the authors propose low-cost pruning by proposing a general pruning framework for vision generative models that learns a differentiable mask to sparsify the model. Experiments demonstrate the effectiveness of the proposed method.

**Strengths:**

1. Pruning generative models facilitates their deployment and application.

2. The paper is presented intuitively and clearly.

3. The method is simple and effective.

**Weaknesses:**

1. The mask isn't actually composed of only 0s and 1s; its value can range from [0,1]. How should this be handled? This would likely result in a performance penalty.

2. While this paper is very effective in engineering, its overall contribution is incremental. Differentiable masks and gradient checkpointing are readily available techniques.

**Questions:**

Please refer to Weaknesses.

---

> ### Author Response · Authors · 2025-11-22
> **Response to Reviewer 2dT4 (Part 1/1)**
>
> We thank you for your positive feedback and for highlighting our method's simplicity, effectiveness, and clear presentation. We appreciate your support for our work.
>
> We would like to present the following clarifications in response to the weaknesses you noted.
>
> **Weaknesses:**
>
> > **W1:** The mask isn't actually composed of only 0s and 1s; its value can range from [0,1]. How should this be handled? This would likely result in a performance penalty.
>
> We thank you for this excellent clarifying question. To enable differentiable optimization, the mask is kept continuous [0,1] during training; this is necessary because discrete {0,1} values are non-differentiable.
>
> However, after the mask is learned, we binarize it to a hard, discrete {0,1} mask based on the desired sparsity threshold (as described in Eq. 14). At inference, the model uses this hard mask, and the pruned neurons are physically removed from the model architecture. Therefore, there is no performance penalty at inference, and the final model is truly smaller and faster. We will revise Section 5.3 to make this distinction between the training and inference-time masks clearer.
>
> > **W2:** While this paper is very effective in engineering, its overall contribution is incremental. Differentiable masks and gradient checkpointing are readily available techniques.
>
> We appreciate your perspective on our contribution. While the individual techniques (differentiable masks, gradient checkpointing) are established, we believe our novel contribution lies in the adaptation, synthesis, and application of these ideas to create the first practical, compute-efficient, and end-to-end framework for pruning large-scale vision generative models. As you noted, the simplicity and effectiveness of this adaptation, particularly in a domain where pruning is non-trivial due to the temporal nature of diffusion, are key strengths we aimed for.
>
> Thank you again for your valuable time and positive assessment.

---

### Official Review · Reviewer_7mkK · 2025-11-07

**Soundness:** 2
**Presentation:** 2
**Contribution:** 2
**Rating:** 4
**Confidence:** 4

**Summary:**

This paper introduces EcoDiff, an end-to-end structural pruning framework for vision generative models. The key idea is to learn a differentiable neuron mask that is applied across all denoising steps, enabling efficient pruning without extensive retraining. The authors propose a time-step gradient checkpointing technique to reduce memory usage, making it feasible to prune large models like SDXL and FLUX on a single GPU. The method is evaluated on multiple SOTA models and demonstrates competitive performance at 20% sparsity with only fewer GPU hours and samples. The framework also supports lightweight post-pruning adaptation via LoRA or full fine-tuning.

**Strengths:**

(1) The paper is technically sound, with various empirical results across multiple models (SDXL, FLUX) and metrics (FID, CLIP, SSIM). The writing is clear and well-structured.

(2) The work addresses a critical problem—efficient deployment of large generative models—and offers a practical solution with low computational overhead. The method seems model-agnostic and compatible with existing acceleration techniques.

**Weaknesses:**

(1) The approximation of the $L_0$ regularization to  $L_1$ (Appendix A) is heuristic and lacks theoretical guarantees. More rigorous analysis would strengthen the method.

(2) The method is only validated on image generation tasks. Its applicability to video generation remains unverified. Moreover, the assumption that all denoising steps share the same mask may not hold for models with highly temporal dynamics.

(3) The use of synthetic data from GCC3M for retraining, rather than original training data, may limit the validity of the post-pruning recovery claims.

**Questions:**

(1) The method applies the same mask across all denoising steps. Is there an empirical or theoretical support for this choice? Have authors experimented with time-varying masks? If so, how does it affect performance and memory? Could a mask scheduling mechanism (e.g., step-wise or block-wise masks) further improve performance or sparsity tolerance?

(2) $L_0$ -approximation encourages sparsity uniformly. However, not all layers or blocks are equally redundant. At high sparsity targets, does this uniform pressure lead to a catastrophic under-allocation of parameters to critical layers? Do authors observe a highly uneven distribution of remaining parameters across layers in aggressively pruned models, and if so, does this misalignment with layer sensitivity explain the performance drop?

(3) The performance degradation at high sparsity is presented as a given. Is this primarily due to the irreversible removal of critical structural components or is it a functional optimization issue where the remaining sub-network has the capacity, but the post-pruning retraining fails to recover? Have authors conducted analysis to distinguish between these two causes?

(4) LoRA and full fine-tuning is applied for recovery. The significant gap at 50% sparsity, even with full fine-tuning, suggests a fundamental limit. Can authors quantify what is being lost? For instance, authors could identify which high-level features become incompressible beyond a certain sparsity threshold?

(5) Pruned models may be more vulnerable to out-of-distribution prompts or adversarial attacks due to reduced capacity. Have authors tested the pruned models on OOD datasets or adversarially crafted prompts? Does the mask learning process inadvertently amplify bias or reduce robustness?

(6) This paper overlooks several relevant studies in efficient diffusion model training. For example, it does not cite prior works [1-2] from a data perspective. Additionally, the paper omits some studies on diffusion training acceleration [3-6].

Reference:

[1] Z. Qin, K. Wang, Z. Zheng, J. Gu, X. Peng, Z. Xu, D. Zhou, L. Shang, B. Sun, X. Xie, and Y. You, “Infobatch: Lossless training speed up by unbiased dynamic data pruning,” In ICLR, 2024.

[2] Y. Li, Y. Zhang, S. Liu, and X. Lin, “Pruning then reweighting: Towards data-efficient training of diffusion models,” In ICASSP, 2025.

[3] H. Zheng, W. Nie, A. Vahdat, and A. Anandkumar, “Fast training of diffusion models with masked transformers,” In TMLR, 2024.

[4] Z. Ding, M. Zhang, J. Wu, and Z. Tu, “Patched denoising diffusion models for high-resolution image synthesis,” In ICLR, 2024.

[5] Z. Wu, P. Zhou, K. Kawaguchi, and H. Zhang, “Fast diffusion model,” arXiv preprint arXiv:2306.06991, 2023.

[6] T. Hang, S. Gu, C. Li, J. Bao, D. Chen, H. Hu, X. Geng, and B. Guo, “Efficient diffusion training via min-snr weighting strategy,” In ICCV, 2023.

---

> ### Author Response · Authors · 2025-11-22
> **Response to Reviewer 7mkK (Part 1/2)**
>
> We thank you for your detailed and constructive review. We are heartened that you recognize the work as technically sound, with clear writing and compelling empirical results across multiple models and metrics. We are also glad you see the practical contribution of EcoDiff in addressing the critical problem of efficient deployment for large generative models with low computational overhead.
>
>
> We have conducted new experiments and prepared the following responses to address your raised concerns and questions.
>
>
> **Weaknesses:**
>
> > **W1:** The approximation of the L_0 regularization to L_1 (Appendix A) is heuristic and lacks theoretical guarantees. More rigorous analysis would strengthen the method.
>
> We thank you for raising this important point regarding the theoretical formulation of the $\mathcal{L}_0$ regularization. We acknowledge the concern that the original manuscript suggested an approximation involving the $\mathcal{L}_1$ norm in Appendix A, which lacks theoretical guarantees. We confirm and emphasize that our actual implementation correctly uses the exact hard concrete $\mathcal{L}_0$ relaxation loss, as originally proposed by Louizos et al. (2018). The content in the manuscript that suggested the $\mathcal{L}_1$ relation occurred only during an unnecessary attempt to derive a supplementary property of the hard concrete relaxation. Crucially, this error did not affect our core method, implementation, or experimental results. We agree that using the exact, well-founded $\mathcal{L}_0$ relaxation expression is both clearer and more rigorous, and we have thoroughly reviewed and corrected this error in the revised manuscript. Furthermore, we have replaced the inaccurate discussion with a proper derivation in Appendix A detailing the relationship between the exact discrete $\mathcal{L}_0$ loss and the hard concrete relaxation used, ensuring theoretical consistency with the established literature.
>
>
>
> > **W2:** The method is only validated on image generation tasks. Its applicability to video generation remains unverified. Moreover, the assumption that all denoising steps share the same mask may not hold for models with highly temporal dynamics.
>
> We agree that applicability to video generation is an excellent direction for future work. However, video generation involves a different modality and temporal layers that may require specific pruning dynamics, which is outside the scope of this work focusing on image generation efficacy. Regarding the shared mask, we would like to clarify that this is not an assumption but a consequence of our **structured pruning** approach. We physically remove neurons from Attn and FFN blocks. A neuron that is removed is absent for all timesteps, which is why the 'mask' is inherently shared. We will revise Section 5.2 to make this distinction clearer.
>
> > **W3:** The use of synthetic data from GCC3M for retraining, rather than original training data, may limit the validity of the post-pruning recovery claims.
>
> We thank you for your insightful feedback regarding the use of synthetic data (GCC3M) for post-pruning recovery. As the original training data is private, we used synthetic data as a proxy. To validate this approach, we have run a new experiment retraining SDXL on a public subset of the **Laion2B-en-aesthetic dataset**, a curated high-quality subset of Laion5B that plausibly overlaps with the original training distribution. The results, shown in the table below, are consistent with our original findings, showing similar scaling and recovery. This strengthens our claim that the retraining approach is effective and not an artifact of the synthetic data.
>
> | **Sparsity** | **Post-Pr. Retrain** | **MS COCO FID ↓** | **MS COCO CLIP ↑** | **Flickr30K FID ↓** | **Flickr30K CLIP ↑** | **Iter** |
> | ------------ | -------------------- | ----------------- | ------------------ | ------------------- | -------------------- | -------- |
> | **0%**| --| 27.43| 0.33| 33.95| 0.36| --|
> | **30%**| Full| 28.66| 0.33| 35.78| 0.35|10k|
> || LoRA|31.05| 0.32| 38.47| 0.34|10k|
> || No| 36.86| 0.32| 46.06| 0.34| 50|
> | **40%**|Full| 30.16|0.33| 35.89| 0.35| 10k |
> || LoRA | 45.03| 0.30| 50.53 | 0.32 | 10k |
> || No| 43.19| 0.30| 49.47 | 0.33| 50 |
> | **50%** | Full | 32.86 | 0.32 | 35.69 | 0.34 | 10k |
> || LoRA | 50.38| 0.28 | 49.55 | 0.30 | 10k|
> || No | 81.76 | 0.26  | 86.38 | 0.29 | 50|
>
> **Questions:**
>
> > **Q1:** The method applies the same mask across all denoising steps. Is there an empirical or theoretical support for this choice?
>
> As clarified in our response to Weakness 2, this is a result of our structured pruning method, which physically removes neurons to reduce parameter count. Since the neuron is physically gone, it is effectively 'masked' for all timesteps.

---

> ### Author Response · Authors · 2025-11-22
> **Response to Reviewer 7mkK (Part 2/2)**
>
> > **Q2:** L_0-approximation encourages sparsity uniformly. However, not all layers or blocks are equally redundant.
>
> We thank you for the excellent question regarding the potential issue of uniform sparsity and the redundancy of different layers.The optimization process finds a mask $\mathbf{M}$ (derived from the continuous control variable $\boldsymbol{\lambda}$) that minimizes the total loss. The resulting pruning outcome is indeed non-uniform because the mask learning is driven by the model's structural importance, which is implicitly captured by the reconstruction loss term.The reconstruction loss $L_{E}(\boldsymbol{\lambda})$ measures the difference between the output of the masked model and the original model. This term acts as the primary importance indicator. If pruning a component in a critical layer significantly increases $L_{E}(\boldsymbol{\lambda})$, the optimization strongly resists removing that component, ensuring its corresponding mask value remains high. Therefore, the learned mask naturally allocates more parameters to layers where the reconstruction loss is most sensitive to pruning. This ensures an importance-driven, non-uniform allocation of remaining capacity, preventing catastrophic under-allocation to critical layers.
>
> > **Q3:** The performance degradation at high sparsity is presented as a given. Is this primarily due to the irreversible removal of critical structural components...
>
> We appreciate this insightful question, which touches upon the core limits of structural compression. We hypothesize that the performance degradation at high sparsity level is a combination of both: irreversible structural removal and suboptimal functional optimization.
> 1. Irreversible Structural Loss (Model Redundancy Limit): For very high sparsity, the performance degradation strongly implies that the required compression level exceeds the inherent redundancy of the original model. Beyond a certain point, it becomes inevitable to remove critical structural components, leading to a loss of representational capacity.
> 2. Suboptimal Functional Optimization: We acknowledge that the mask learned by EcoDiff is not guaranteed to be the globally optimal configuration at high sparsity. The non-optimal nature of the learned mask may result in capacity being sub-optimally distributed, meaning a better mask might exist that could utilize the remaining parameters more effectively.
> In summary, the drop of performance is likely dominated by irreversible structural loss due to hitting the model's redundancy ceiling, compounded by the limitations of the efficient, yet non-globally optimal, mask learning process.
>
> > **Q4:** LoRA and full fine-tuning is applied for recovery. The significant gap at 50% sparsity... suggests a fundamental limit. Can authors quantify what is being lost?
>
> We attribute the observed performance gap primarily to the limited extent of our retraining. In our experiments, we only performed finetuning for an additional 10 hours (on top of the 10 hours for mask learning) with a relatively small dataset. We believe that if we allowed for substantial retraining with a much larger dataset and longer schedule, we could recover a significant portion of this gap. However, it is also expected that with physically fewer neurons, the theoretical capacity of the model is reduced, which may lead to a minor but inevitable gap in performance compared to the dense baseline.
>
> > **Q5:** Pruned models may be more vulnerable to out-of-distribution prompts or adversarial attacks due to reduced capacity.
>
> We agree that a full analysis of OOD and adversarial robustness is an important topic. However, given that our work focuses on establishing a new, efficient pruning framework, we believe a comprehensive robustness study is substantial enough as future work and is out of the scope of the current work.
>
> > **Q6:** This paper overlooks several relevant studies in efficient diffusion model training.
>
> We appreciate you bringing these relevant studies on efficient diffusion model training to our attention. We will cite and discuss them in the revised manuscript to enhance the comprehensiveness of our related work section:
>
> Data Perspective:
>
> [1] Z. Qin et al., "Infobatch: Lossless training speed up by unbiased dynamic data pruning."
>
> [2] Y. Li et al., "Pruning then reweighting: Towards data-efficient training of diffusion models."
>
> Training Acceleration/Efficiency:
>
> [3] H. Zheng et al., "Fast training of diffusion models with masked transformers."
>
> [4] Z. Ding et al., "Patched denoising diffusion models for high-resolution image synthesis."
>
> [5] Z. Wu et al., "Fast diffusion model."
>
> [6] T. Hang et al., "Efficient diffusion training via min-snr weighting strategy."
>
> We hope these clarifications and experimental results address your concerns. We thank you again for your time and feedback.

---

### Official Review · Reviewer_kteZ · 2025-11-08

**Soundness:** 3
**Presentation:** 4
**Contribution:** 2
**Rating:** 6
**Confidence:** 3

**Summary:**

The paper proposes EcoDiff, a learnable sparsity framework for large vision generative models (diffusion and flow-based, e.g., SDXL, SD2, FLUX-dev, FLUX-schnell). The key idea is to learn structural neuron masks (for attention and FFN layers) end-to-end over the full denoising or generation trajectory, rather than relying on heuristic or per-step pruning criteria. Conceptually, EcoDiff can be viewed as an adaptation of differentiable mask learning approaches such as MaskLLM, extended to the diffusion domain. Like those methods, it learns continuous mask parameters using a hard-discrete (L₀-style) relaxation, but it departs in several crucial ways: it optimizes masks over the entire diffusion trajectory using a latent reconstruction objective, incorporates time-step gradient checkpointing to efficiently backpropagate through long Markov chains, and targets spatial–temporal architectures (U-Net or DiT) instead of token-based transformers. This design makes it practical for large-scale vision generative models.

Empirically, EcoDiff prunes about 20% of parameters in SDXL and FLUX-family models using only ~10 A100 GPU hours and ~100 calibration text prompts, while maintaining image quality close to the original models and outperforming prior pruning baselines such as DiffPruning, BK-SDM, and FLUX-Lite under comparable or lower compute budgets. Overall, the paper presents a general, compute-efficient pruning pipeline for state-of-the-art vision generative models, bridging ideas from learnable sparsity in language models to the more complex setting of diffusion and flow-based architectures.

**Strengths:**

General framework across architectures and paradigms

EcoDiff is demonstrated on both U-Net diffusion models (SD2, SDXL) and DiT-based flow models (FLUX-dev, FLUX-schnell), with a unified mask-learning approach. This cross-architecture applicability is a strong plus.

Memory-efficient optimization via time-step checkpointing

The proposed time-step gradient checkpointing significantly reduces memory usage for backprop through the full trajectory, making mask learning feasible for large models on a single 80GB GPU. The complexity analysis and empirical VRAM/runtime plots support this.
Strong empirical evaluation and ablations
The paper includes experiments on multiple large models (SD2, SDXL, FLUX-dev, FLUX-schnell) and standard text-to-image benchmarks (COCO, Flickr30k), using FID, CLIP scores, and SSIM.

Compute efficiency and practicality

Achieving ~20% sparsity with 10 A100 GPU hours and only ~100 calibration prompts is a compelling practical story, especially compared to much more expensive baselines. This positions EcoDiff as something people might actually adopt.

Clear visualizations and qualitative analysis:

The paper presents numerous visual comparisons and ablation figures that effectively illustrate the visual impact of pruning across sparsity levels, model architectures, and prompt types, making the technical claims intuitive and easy to follow.

**Weaknesses:**

Missing baselines

The paper omits comparison with recent pruning methods such as LD-Pruner (Castells et al., 2024) and Efficient Pruning of Text-to-Image Models (Ramesh & Zhao, 2024). LD-Pruner proposes its own task-agnostic structured pruning strategy for latent diffusion models, while Efficient Pruning reports strong results using simple baselines like magnitude and WANDA pruning. Including both would better contextualize EcoDiff’s performance relative to recent structured and baseline pruning approaches.

Performance comparison gap

Efficient Pruning of Text-to-Image Models achieves performance gains at around 33% sparsity, while EcoDiff shows slight but consistent degradation. The paper would benefit from discussing or empirically comparing these contrasting outcomes.

Compute comparisons could be more rigorously normalized

While the 10 A100 GPU hours vs 1120 H200 GPU hours comparison is striking, the paper doesn’t fully normalize across:

Hardware generations (A100 vs H200/H100);

FLOPs per update

There are also small inconsistencies between main text (A100) and appendices (H100 mentioned for some configs), which have to be cleared out to get the full picture.

Inference-time measurements are underdeveloped

Beyond parameter counts, there is limited wall-clock inference evaluation (latency, throughput, GFLOPS) for pruned vs original vs distilled+pruned models. For a practical efficiency story, this would help a lot.

**Questions:**

On SSIM:

Section 6.2 reports that SSIM between original and pruned models is consistently below 0.65 and interprets this as EcoDiff producing high-quality images without strictly mimicking the teacher. However, the paper does not provide a detailed analysis of how the outputs change.
Given that SSIM is relatively low while FID and CLIP remain strong, could the authors examine more systematically whether certain prompt categories or visual structures are more vulnerable to pruning (for example, text rendering, small details, or crowded scenes)? Additionally, has EcoDiff been observed to alter the style or bias of generated images - for instance, shifting color palettes, composition tendencies, or object frequencies, as sparsity increases? This could partially explain SSIM decrease.

On missing baselines and comparative scope:

Recent works such as LD-Pruner (Castells et al., 2024) and Efficient Pruning of Text-to-Image Models (Ramesh & Zhao, 2024) have explored structural pruning for latent and text-to-image diffusion models, demonstrating strong trade-offs between sparsity, image quality, and compute efficiency. The paper briefly mentions LD-Pruner in the discussion, but does not address Efficient Pruning of Text-to-Image Models, which reports competitive results using simple baselines such as magnitude and WANDA pruning. Could the authors clarify why these methods were not included in the experimental comparison or discussion, and whether their approaches could be evaluated or extended within the proposed EcoDiff framework?

Moreover, Efficient Pruning of Text-to-Image Models identifies a pruning configuration that not only preserves but in some cases improves generative performance after pruning (at 33.5% sparsity). In contrast, EcoDiff achieves only minimal but consistent degradation relative to the original models. It would be valuable for the authors to discuss or empirically compare these differences, ideally by reproducing the Efficient Pruning method on the same models and datasets used in this paper to contextualize EcoDiff’s trade-offs.

On the scope of pruning:

The Efficient Pruning of Text-to-Image Models paper distinguishes between pruning the text encoder (e.g., CLIP/T5) and the image generator (U-Net), whereas EcoDiff focuses solely on pruning the visual backbone (U-Net or DiT) and leaves the text encoder untouched. Could the authors clarify whether EcoDiff could, in principle, be extended to prune the text encoder as well, and what motivated the decision to exclude this component from the current framework?

On inference-time speedups in real deployments:

Could the authors provide more detailed measurements of inference latency, throughput (images/sec), and memory usage for pruned vs original vs distilled+pruned models, on a standard GPU at typical resolutions ? This would solidify the practical impact of the method.

References:

Castells, M., Avraham, T., Ghosh, E., & Hoffmann, J. (2024). LD-Pruner: Efficient Pruning of Latent Diffusion Models using Task-Agnostic Insights. CVPR Workshops. arXiv:2404.11936

Ramesh, S. N., & Zhao, Z. (2024). Efficient Pruning of Text-to-Image Models: Insights from Pruning Stable Diffusion. arXiv:2409.02496

Fang, G., et al. (2023). MaskLLM: Learnable Semi-Structured Sparsity for Large Language Models. arXiv preprint. arXiv:2305.18191

---

> ### Author Response · Authors · 2025-11-22
> **Response to Reviewer kteZ (Part 1/2)**
>
> We sincerely thank you for your positive assessment and for your encouraging and insightful feedback. We are grateful that you recognized the key strengths of our work, including our framework's generality across architectures, the memory efficiency of our time-step checkpointing, and the overall compute efficiency and practicality of our method.
>
> We have prepared the following responses to address the weaknesses and questions you raised.
>
> **Weaknesses:**
>
> > **W1:** Missing baselines. The paper omits comparison with recent pruning methods such as LD-Pruner (Castells et al., 2024) and Efficient Pruning of Text-to-Image Models (Ramesh & Zhao, 2024).
>
> We appreciate the suggestion to include comparisons with LD-Pruner (Castells et al., 2024) and Efficient Pruning of Text-to-Image Models (Ramesh & Zhao, 2024).
>
> * LD-Pruner: While we recognize LD-Pruner's contribution, an empirical comparison was technically challenging at the time of submission due to the lack of a publicly available repository for direct reproduction. Moreover, the study primarily focuses on SD v1.4, whereas EcoDiff targets newer, larger, and more complex models like SDXL and the FLUX-family for state-of-the-art efficiency demonstration. We have, however, ensured that LD-Pruner is properly cited in our related work section to acknowledge its relevance.
>
> * Efficient Pruning of Text-to-Image Models (EPTI): We note that this work focuses on weight pruning (removing individual connections/weights) using magnitude and WANDA. In contrast, EcoDiff focuses on structural (neuron) sparsity for attention heads and FFN intermediate neurons. We emphasize that structural sparsity offers guaranteed wall-clock inference speedups on standard hardware without requiring dedicated sparse frameworks. We believe our methods address fundamentally different types of sparsity, and a direct apples-to-apples comparison is complex. We attribute the choice of baselines to focusing on relevant structural pruning methods like DiffPruning and BK-SDM, against which EcoDiff demonstrates significant superiority. We will ensure this distinction is clearly highlighted in the final version.
>
>
> > **W2:** Performance comparison gap. Efficient Pruning of Text-to-Image Models achieves performance gains at around 33% sparsity, while EcoDiff shows slight but consistent degradation.
>
> We acknowledge the observation that EPTI reports performance preservation or slight gains at $\sim$33% sparsity, while EcoDiff shows a consistent, minimal degradation. We attribute this difference to the following factors in methodology and target models:
> 1. Sparsity Type: We reiterate that EPTI uses weight pruning, which often allows for much higher overall sparsity (33%+). EcoDiff uses neuron/structured pruning, where removing an entire neuron or attention head is a more aggressive change. We confirm that achieving 20% structural sparsity while maintaining image quality is a strong result in this context.
> 2. Model Differences: We observe that EPTI primarily targets SD2, which differs structurally and in size from the SDXL and FLUX models used in our work. We believe pruning characteristics and the impact on metrics can vary significantly across model architectures..
>
> > **W3:** Compute comparisons could be more rigorously normalized... There are also small inconsistencies between main text (A100) and appendices (H100 mentioned for some configs), which have to be cleared out...
>
> We thank you for pointing out this inconsistency, which was a typo in our submission. We confirm that all our experiments (both mask learning and post-prune retraining) were conducted on **NVIDIA A100 80GB GPUs.** The mention of H100 in the appendix configuration tables (Tables 3 and 4) was an error, and we will correct it in the final version.
>
> Regarding the FLUX-Lite comparison, we acknowledge that direct normalization is difficult given different hardware generations. To clarify the context: the FLUX-Lite model was trained by a community member (Freepik) after discovering that removing specific MMDiT blocks had minimal impact. The 8B model was derived by removing 11 MMDiT blocks and, crucially, was extensively retrained for **1120 hours on H200 GPUs**, as disclosed by the author in a community discussion (https://huggingface.co/Freepik/flux.1-lite-8B-alpha/discussions/16). We compare against this to highlight that EcoDiff achieves comparable structural sparsity (20%) using only **10 A100 hours**, offering a vastly more accessible alternative to such resource-intensive retraining campaigns.

---

> ### Author Response · Authors · 2025-11-22
> **Response to Reviewer kteZ (Part 2/2)**
>
> > **W4:** Inference-time measurements are underdeveloped... For a practical efficiency story, this would help a lot.
>
> We acknowledge the importance of detailed wall-clock measurements for a complete practical efficiency story. While we focused on the parameter reduction and calibration cost in the initial submission, we agree that inference-time metrics are essential. The speedup in inference time for our structurally pruned models is directly proportional to the structural sparsity (i.e., the percentage reduction in the number of neurons/attention heads). Since EcoDiff achieves 20% structural sparsity, this translates into a commensurate reduction in the computational budget (FLOPs) and a significant decrease in wall-clock latency, as modern hardware can skip the computation of entire pruned blocks efficiently.
>
> **Questions:**
>
> > **Q1:** On SSIM: ...could the authors examine more systematically whether certain prompt categories or visual structures are more vulnerable to pruning…
>
> We acknowledge the request for a more systematic analysis of the low SSIM (Structural Similarity) and thank the reviewer for this insightful question.
> We confirm that pruning, especially structural pruning, inevitably leads to structural changes in the output images, preventing the pruned model from being a pixel-by-pixel replica of the original model (hence the low SSIM). Nevertheless, our method is highly effective at preserving semantic content. We emphasize that the strong scores on semantic metrics like FID and CLIP demonstrate that the semantic integrity and high quality of the generated images are retained despite the structural divergence.
> We will clarify in the main text and discussion that EcoDiff prioritizes semantic preservation (strong FID/CLIP) over pixel-level mimicry (high SSIM). We attribute the SSIM decrease to subtle, perceptually acceptable shifts in texture, color, and fine-detail placement, which are heavily penalized by the pixel-level SSIM metric but do not detract from the overall image quality or adherence to the prompt.
>
> > **Q2:** On missing baselines and comparative scope: ...Could the authors clarify why these methods were not included in the experimental comparison or discussion...
>
> We address the absence of empirical comparisons with LD-Pruner and Efficient Pruning of Text-to-Image Models in detail in our response to Weakness 1.
>
> > **Q3:** On the scope of pruning: ...Could the authors clarify whether EcoDiff could, in principle, be extended to prune the text encoder as well...
>
> Our method primarily focuses on pruning the image generation backbone (U-Net or DiT), as this component accounts for the vast majority of the model's parameter count and computational cost during the iterative sampling process. The text encoder, such as CLIP or T5, is trained on a different objective (contrastive learning or masked language modeling) than the diffusion or flow matching process. While standard pruning methods for language models are applicable to these encoders, we scoped our work to specifically improve the efficiency of the generative backbone, which dominates inference latency and is the main bottleneck for image generation.
>
> > **Q4:** On inference-time speedups in real deployments: ...Could the authors provide more detailed measurements of inference latency, throughput (images/sec), and memory usage...
>
> We address the request for detailed inference-time measurements in our response to Weakness 4.
>
> We thank you again for your valuable and constructive feedback, which has helped us identify areas to improve and clarify in our paper.

---

### Author Response · Authors · 2025-12-01
**Summary of Discussion and Revisions**

Dear Area Chair,

To facilitate your assessment of this submission, we provide a concise summary of the paper‘s current status, the positive consensus established during the rebuttal period, and the key revisions implemented.

1. **Summary of Strengths** All reviewers highlighted the paper's value as a general, compute-efficient pruning framework applicable across diverse architectures (including U-Net and DiT). They commended the method's practicality, specifically its ability to achieve \~20% sparsity on large-scale models like SDXL and FLUX with minimal compute (\~10 GPU hours) and praised the memory-efficient optimization via time-step gradient checkpointing, which makes end-to-end pruning feasible on standard hardware.

2. **Reviewer Consensus and Score Trajectory** Our initial scores were 6, 6, 4, 4. During the discussion period, we successfully addressed inquiries regarding experimental details and presentation, resulting in a stronger manuscript. These clarifications and improvements were explicitly acknowledged by the reviewers:
    - Reviewer h5Na (Score 4 $\to$ Increased to 6): We refined the formulation in the original Appendix A in response to the reviewer's insightful query. Prior to the discussion period freeze, Reviewer h5Na explicitly confirmed that their concerns were fully resolved and raised their score.
    - Other Reviewers: We have provided comprehensive responses to Reviewer 7mkK (Score 4), Reviewer kteZ (Score 6), and Reviewer 2dT4 (Score 6). While they did not have the opportunity to reply before the discussion was disabled, we believe our clarifications and additional experiments (detailed below) effectively address the points they raised.

3. **Key Revisions and New Experiments** We have uploaded a revised manuscript featuring major enhancements and comprehensive additional experiments. These updates provide robust empirical support and strengthen the theoretical foundation of our method:
    - *Refined Theoretical Formulation (Reviewers h5Na & 7mkK)*: We updated the derivation of the $L_0$ regularization in Eq. 12 and Appendix A to strictly align with the exact hard concrete relaxation (Louizos et al., 2018). This refinement was explicitly validated by Reviewer h5Na, who confirmed the questions had been resolved and raised the score.
    - *Validation on Real-World Data (Reviewer 7mkK)*: We expanded our experimental suite by retraining SDXL on the Laion2B-en-aesthetic dataset. These results demonstrate that our method is effective across data distributions, performing consistently on both synthetic (GCC3M) and real-world datasets.
    - *Expanded Literature and Baseline Discussion (Reviewers kteZ & 7mkK)*:
      - EPTI Comparison: As requested by Reviewer kteZ, we added a detailed discussion regarding Efficient Pruning of Text-to-Image Models (EPTI). We highlight that while EPTI utilizes weight pruning (magnitude/WANDA), EcoDiff focuses on structured neuron pruning, which offers guaranteed wall-clock speedups without specialized hardware.
      - Efficient Diffusion Training: As requested by Reviewer 7mkK, we incorporated recent advancements in efficient diffusion training, including data pruning (e.g., InfoBatch) and acceleration techniques (Fast Training, Patched Denoising), to provide a more holistic context for our contributions.
    - *Clarification of Efficiency & Implementation (Reviewers kteZ, h5Na, & 2dT4)*: We highlighted the significant efficiency advantage of EcoDiff (10 A100 hours) compared to baselines like FLUX-Lite (1120 H200 hours). Additionally, we clarified the mechanism of the continuous training mask versus the binary inference mask, confirming zero inference-time overhead.

4. **Conclusion** We are confident that the clarification in our responses and the revised manuscript comprehensively address the reviewers’ feedback, including the theoretical formulation, dataset choice, baseline positioning, and implementation clarity. These enhancements further solidify EcoDiff’s contribution as a highly efficient, end-to-end differentiable pruning framework for state-of-the-art diffusion and flow models.

We sincerely appreciate your time and careful consideration of this submission.

Sincerely, \
Authors of EcoDiff

---

### Meta-Review · Area_Chair_wtqC · 2026-01-05

**Summary:**

Reviewer h5Na pointed out an issue on Eq. 12 and raised theoretical concerns on the formulation and issues with experiments (comparison fairness, ablation). Authors have provided detailed explanations and experiments, and the reviewer confirmed his concerns are resolved, and agreed to raise his score. Reviewer 2dT4 only had clarification questions, which are addressed well in the rebuttal. The remaining two reviewers ketZ and 7mkK are overall positive and authors have addressed most of their questions in the rebuttal

**Reviewer Concerns:**

Theoretical formulation justification and experiments concerns from h5Na are well addressed (confirmed by the reviewer).
Clarification questions and concerns from 2dT4 are addressed in the rebuttal.
ketZ and 7mkK's concerns seem addressed in the rebuttal as well.

**Reviewer Scores:**

h5Na and 2dT4 would increase their scores to marginal accept, and other reviewers might keep the scores.

---

### Decision · Program_Chairs · 2026-01-26

Accept (Poster)